# Distributional Autoencoders Know the Score

**Andrej Leban**
Department of Statistics,
University of Michigan,
Ann Arbor, MI, United States
`leban@umich.edu` *

## Abstract

The Distributional Principal Autoencoder (DPA) combines distributionally correct reconstruction with principal-component-like interpretability of the encodings. In this work, we provide exact theoretical guarantees on both fronts. First, we derive a closed-form relation linking each optimal level-set geometry to the data-distribution score. This result explains DPA's empirical ability to disentangle factors of variation of the data, as well as allows the score to be recovered directly from samples. When the data follows the Boltzmann distribution, we demonstrate that this relation yields an approximation of the minimum free-energy path for the Müller–Brown potential in a single fit. Second, we prove that if the data lies on a manifold that can be approximated by the encoder, latent components beyond the manifold dimension are conditionally independent of the data distribution — carrying no additional information — and thus reveal the intrinsic dimension. Together, these results show that a single model can learn the data distribution and its intrinsic dimension with exact guarantees simultaneously, unifying two longstanding goals of unsupervised learning.

## 1 Introduction

The Distributional Principal Autoencoder (DPA) is an autoencoder variant recently introduced in Shen and Meinshausen [18] and related to other approaches [19; 20] built on the *energy score* [10]. Because the decoder is trained to match the conditional law given a code, DPA guarantees *distributionally correct* reconstruction for all points mapped to the same code. The encoder, in turn, minimizes the residual variability given this code. Optimizing several latent widths simultaneously induces a principal–components-like ordering while retaining an expressive nonlinear mapping. Although the original paper demonstrated strong performance on high-dimensional data, a precise theoretical explanation of the behavior remained open.

We close this gap by proving strong properties for both *data distribution learning* and *dimensionality reduction* aspects which hold *simultaneously* in DPA — two objectives that are almost always at odds in unsupervised learning. First, we derive a closed-form *score–geometry identity* that links the normal components of each optimal level set to the data score $\nabla \log P_{data}(x)$. Second, we prove that if the data lie on a manifold that is parameterizable (or best approximable) by the encoder, then coordinates beyond the manifold dimension become conditionally independent of the data, thereby revealing the intrinsic dimension. Thus, we extend the analogy to PCA made in the original DPA work by proving that, instead of finding principal linear subspaces, DPA learns *nonlinear manifolds* shaped by the data density, with a clear, testable dimensionality criterion — *conditional independence*. The results hold for *any* combination of ambient dimension $p$ and latent dimension $k$; in examples, we focus on lower-dimensional ones in order to build geometric intuition.

---

*Code available at: [github.com/andleb/DistributionalAutoencodersScore](github.com/andleb/DistributionalAutoencodersScore)

39th Conference on Neural Information Processing Systems (NeurIPS 2025).

The overall organization is as follows: Section 2 introduces the score–geometry identity and its consequences, Section 3 establishes conditional independence of extraneous latents and related results on intrinsic dimensionality. Section 4 illustrates both findings on examples, and demonstrates that the same score alignment enables a *single-fit* recovery of the minimum free-energy path (MFEP) for the Müller–Brown potential—a longstanding benchmark in molecular simulations. This connection underscores the work's relevance to science: encodings that approximate the MFEP can significantly speed up expensive molecular dynamics simulations by identifying reaction pathways directly from samples. Section 5 summarizes prior work relevant to both main results, and Section 6 concludes with broader implications.

## 2 Optimal encoder level sets align with the data score

### 2.1 Background and preliminaries

We denote the data by $X \sim P_{\text{data}}$ supported on $\mathcal{X}$. DPA couples a deterministic encoder $e : \mathbb{R}^p \to \mathbb{R}^k$ with a stochastic decoder $d : \mathbb{R}^k \to \mathbb{R}^p$. Optimal encoder/decoder(s) will typically be denoted by $*$. The decoder is not trained to predict a single point; using the *energy score*, it learns to produce the entire conditional distribution of the data when the encoding value (or "code") is fixed; this distribution is defined as the *Oracle Reconstructed Distribution* (ORD). Thus, points that share the same code form a level set in the data space, and samples from the decoder are draws from the data distribution restricted to that set. This is the sense in which DPA performs distributionally correct reconstruction.

The "principal" part comes from optimizing all dimensions of the encoding (up to some $K$) simultaneously. The $k$-th term of the objective asks how much variability remains if we restrict ourselves to the first $k$ coordinates of $e(X)$; minimizing these terms together orders the coordinates by how much residual variability they remove, analogous to principal components but with a flexible, nonlinear mapping and distributional reconstructions. We formalize these notions below; a compact notation summary appears in App. A.1.

**Definition 2.1 (Oracle reconstructed distribution – ORD, Definition 1 in [18] )**
*For a given encoder $e(\cdot) : \mathbb{R}^p \to \mathbb{R}^k$ and a given sample $x \in \mathbb{R}^p \sim P_{data}$, the oracle reconstructed distribution, denoted by $P_{e,x}^*$, is defined as the conditional distribution of $X$, given that its embedding $e(X)$ matches the embedding $e(x)$ of $x$:*

$$(X \mid e(X) = e(x)) \sim P_{e,x}^* \tag{1}$$

In other words, the ORD is the distribution of the data on the *level set* of the encoder.

**Definition 2.2 (DPA encoder optimization objective, Eq. 4 in [18])**
*The optimal DPA encoder seeks to minimize the expected variability in the induced oracle reconstructed distribution:*

$$e^* \in \underset{e}{\arg\min} \, \mathbb{E}_{X \sim P_{data}} \left[ \mathbb{E}_{Y,Y' \overset{iid}{\sim} P_{e,X}^*} \left[ \|Y - Y'\|^\beta \right] \right], \tag{2}$$

*with $\beta$ a hyperparameter, and the norm taken to be the Euclidean norm in $\mathbb{R}^p$.*

We will denote the **level set** of an optimal encoder as:

$$L_{e^*(X)} \overset{\Delta}{=} \{y : e^*(y) = e^*(X)\}, \tag{3}$$

### 2.2 Results

**Assumption 2.3 (Global Assumptions)**
*We assume the following throughout this section:*

1) *The data density $P_{data}$ is $\mathcal{C}^1$ and integrable.*

2) *The encoder $e$ is $\mathcal{C}^1$ and Lipschitz.*

The following assumption is a *local one*: it is only required where noted, and in case it does not hold, the statements that require it are *silent* for the violating set.

**Assumption 2.4 (Local Rank)**
*The (optimal) encoder's Jacobian matrix evaluated at $y$ — $D_e^*(y)$ — has full row rank $k$ for almost every $y$ on $L_{e^*(X)}$.*

The first lemma introduces a general relationship governing the expected $\beta$-th power of the distance on a level set of an optimal encoder:

**Lemma 2.5 (General integral balance for an optimal encoder)**
*For any $\beta > 0$, assume Assumptions 2.3 and define:*

$$J_\beta\left(y; X, \eta\right) = \int \nabla_{y'} \cdot \left[ D_{e^*}^\top(y') \, \|y - y'\|^\beta \, P_{data}(y')(\eta(y') - \eta(X)) \right] \delta(e^*(y') - e^*(X)) \, dy',$$

$$S(X, \eta) = \int \nabla_z \cdot \left[ D_{e^*}^\top(z) P_{data}(z)(\eta(z) - \eta(X)) \right] \delta(e^*(z) - e^*(X)) \, dz, \tag{4}$$

*where $\eta$ is a perturbation (function), and $\delta$ is the Dirac delta distribution. Next, define the **level-set mass**:*

$$Z(X) = \int P_{data}(z) \, \delta(e(z) - e(X)) \, dz. \tag{5}$$

*Then, for almost every sample $X \sim P_{\text{data}}$ whose level set $\mathcal{L}_{e^*(X)}$ satisfies Assumption 2.4, and any $\eta$ in the same function class as $e$, we have:*

$$\mathbb{E}_{Y \sim P_{e^*,X}^*}\left[ J_\beta\left(Y; X, \eta\right) \right] = \frac{S(X, \eta)}{Z(X)} \, \mathbb{E}_{Y,Y' \overset{iid}{\sim} P_{e^*,X}^*}\left[ \|Y - Y'\|^\beta \right] \tag{6}$$

*provided the quantities in Eqs. 4 are finite and $Z(X) > 0$.*

We can interpret this relation as a balancing act: the terms $J_\beta$ and $S$ represent a "push" from the data density to deform the level set, while the "variance" term on the RHS "pulls" it closer as it needs to be minimized on the level set. While the above relationship holds for any $\beta > 0$ and is ultimately about the *expectations* on a level set, a more striking result occurs when we set $\beta = 2$, which yields a *pointwise* balance equation *exactly* relating the encoder geometry to the data score:

**Theorem 2.6 (When $\beta = 2$, the optimal encoder's level sets align with the data score)**
*Fix $\beta = 2$ and assume Assumptions 2.3. Then, for almost every sample $X \sim P_{\text{data}}$ whose level set $\mathcal{L}_{e^*(X)}$ satisfies Assumption 2.4, the following balance equation holds for almost every $y \in \mathcal{L}_{e^*(X)}$:*

$$\boxed{\frac{2\big(y - c(X)\big)}{\dfrac{V(X)}{Z(X)} - \|y - c(X)\|^2} \, D_{e^*}^\top(y) \;=\; s_{\text{data}}(y) \, D_{e^*}^\top(y),} \tag{7}$$

*where $s_{\text{data}}(y) := \nabla_y \log P_{\text{data}}(y)$ is the Stein score, whenever the following quantities: the level-set **center-of-mass**:*

$$c(X) = \frac{1}{Z(X)} \int y \, P_{data}(y) \, \delta(e(y) - e(X)) \, dy, \tag{8}$$

*and the level-set **variance**:*

$$V(X) = \int \|y - c(X)\|^2 \, P_{data}(y) \, \delta(e(y) - e(X)) \, dy. \tag{9}$$

*are finite and $Z(X) > 0$.*

The proofs of the lemma and the theorem are based on deriving balance equations from the first variation of the encoder's optimization objective (Eq. 2) and are presented in Appendix A.2.

Eq. 7 expresses a trade-off: the variance minimization objective (Eq. 2) pulls the level set toward its center of mass $c(X)$, whereas the local data geometry, through the score, pushes back. The balancing factor compares a *global* term, $V(X)/Z(X)$, to a *local* term, $\|y - c(X)\|^2$. For the unprojected radial vector $y - c(X)$ there are three regimes: for small radii it points outward; beyond the critical radius $r_* = \sqrt{V(X)/Z(X)}$ the sign flips and it points inward; the critical shell itself has measure zero on a level set and is excluded from the theorem. After projection to the normal space via $D_{e^*}^\top$, the level sets align these "outward/inward" directions with the data score in a manner that satisfies both the variance minimization and the local distribution constraints.

The projection to the normal space $D_{e^*}^\top$ is natural: the encoding value changes only in directions normal to the level set, so optimal level sets must align those normal directions with the change in density (the score). Furthermore, the normal space has dimension $k$, making explicit the trade-off between dimensionality reduction (from $p$ to $k$) and how strictly alignment with the score can be enforced (i.e., in $k$ of the $p$ dimensions). As shown in Sec. 3, if the data lie on a $k$-dimensional manifold, DPA can, in principle, align with the score in all relevant directions while rendering the remaining $(p - k)$ coordinates extraneous.

As in Shen and Meinshausen [18], an optimal *encoder* is well defined for $\beta = 2$, which suffices for Theorem 2.6. The only subtlety concerns the optimal *decoder uniqueness*: the energy-score is *proper* but not *strictly proper* at $\beta = 2$, so distinct conditional laws can attain the same risk even if only one equals the ORD. In our experiments we did not observe such degeneracy; the learned encodings approximate the data score (Sec. 4).

The requirement that the Jacobian has full row rank almost everywhere on the level set can be violated in practice if, e.g., the encoder is suffering from "mode collapse" and mapping large regions of the input to exactly the same constant. Another cause might be that the neural network is not sufficiently expressive to approximate the manifold, which is assumed *not* to be the case throughout this work. As mentioned, Theorem 2.6 is simply *silent* on those level sets; in our examples we observe well-behaved level sets.

A direct consequence for extrema follows by setting the right-hand side of Eq. 7 to zero:

**Corollary 2.7 (Consequence of Theorem 2.6 for extrema of the data distribution)**
*Adopt all the assumptions of Theorem 2.6. Then, excluding minima where $\|y^* - c(X)\|$ approaches infinity, an optimal encoder's level sets at extrema will either:*

1) *have the center-of-mass $c(X)$ at the extremum: $y^* = c(X)$, or*

2) *have the vector from the extremum to the center-of-mass $(y^* - c(X))$ tangent to $L_{e^*(X)}$.*

The proof is located in Appendix A.2.3. Due to the variance minimization objective, it is typically suboptimal for the same level set to cross multiple maxima of the data distribution. If it does occur, it will often mean that the level set between them is approximately a line segment, as $c(X)$ will lie close to the connecting line (due to the density weighting) and $(y - c(X))$ must be tangent in both.

# 3 On (approximately) parameterizable manifolds, extraneous latents are uninformative

## 3.1 Background and preliminaries

In this section we relate the intrinsic dimension $K$ of the data manifold to encoder coordinates beyond $K$. The decoder is taken to be a stochastic network that takes noise $\epsilon \sim \mathcal{N}(0, I_{(p-k)})$ (by default) as input, with $\epsilon \perp\!\!\!\perp X$, and, by Theorem 1 in Shen and Meinshausen [18]: $d^* (e^*(x), \epsilon) \sim P_{e^*,x}^*$.

Given this, one can consider multiple possible encoder output dimensions $k$, with the remaining input dimensions of $d$ being padded by expanding $\epsilon$: $d\left([e_{1:k}(x), \epsilon_{(k+1):p}]\right) \sim P_{d,e_{1:k}}(X)$. Then, the joint optimization objective across all components is (Eq. 12 in [18]):

$$(e^*, d^*) \in \operatorname*{argmin}_{e,d} \sum_{k=0}^{p} \omega_k \left[ \mathbb{E}_X \mathbb{E}_{Y \sim P_{d,e_{1:k}}(X)} \left[\|X - Y\|^\beta\right] - \frac{1}{2}\mathbb{E}_X \mathbb{E}_{Y,Y' \overset{\text{iid}}{\sim} P_{d,e_{1:k}}(X)} \left[\|Y - Y'\|^\beta\right] \right],$$
(10)

where $\omega_k \in [0, 1]$ are (optional) weights. In this section, we will assume uniform weights: $\frac{1}{p+1}$, and $\beta \in (0, 2)$, which makes the objective a *strictly proper scoring rule* [10], and thus the global optimum *unique*. This is a technical necessity for the proofs; however, using $\beta = 2$ yields comparable empirical results, as demonstrated in Sec. 4.2.

We denote the $k$-th term of the optimization objective 10 as:

$$L_k[e, d] = \mathbb{E}_X \mathbb{E}_{Y \sim P_{d,e_{1:k}}(X)} \left[\|X - Y\|^\beta\right] - \frac{1}{2}\mathbb{E}_X \mathbb{E}_{Y,Y' \overset{\text{iid}}{\sim} P_{d,e_{1:k}}(X)} \left[\|Y - Y'\|^\beta\right]. \quad (11)$$

## 3.2 Results

We will assume the data $X \sim P_{data}$ to lie on a $K$-dimensional manifold, with $K < p$. We first focus on a case where the manifold cannot be exactly parameterized via an encoder, but can be approximated in the *energy score* sense - for instance, a union of manifolds. Throughout, we assume sufficient network expressivity.

**Definition 3.1 ($K'$-parameterizable manifold)**
*A $K$-dimensional manifold is $K'$-parameterizable if it can be approximated in the minimum energy score sense in $K'$-dimensions, that is, for an optimal encoder/decoder pair, the $K'$-term in the loss Eq. 10 is globally the smallest among all terms* and *among all encoder/decoder pairs:*

$$L_{K'}[(e^*, d^*)] = \min_{e,d,k} L_k[e, d] \tag{12}$$

Naturally, we have $K' \geq K$. As $K$ dimensions cannot be fully "captured" by a lower-dimensional mapping, the first – reconstruction term in the loss could always be reduced by considering $K$ dimensions; at optimum, the second term in Eq. 11 equals the reconstruction term (without the $\frac{1}{2}$ factor, cf. Prop. 1 in [18]). Thus we have a contradiction as that cannot be minimal, either, implying that $K' \geq K$.

Next, we show that this definition is compatible with the usual notion of exact parameterizability:

**Proposition 3.2 (An exactly parameterizable manifold is $K'$-parameterizable)**
*Consider the case where the data is located on a $K$-dimensional manifold $\subset \mathbb{R}^p$, which is exactly parameterizable by some function $\mathbb{R}^p \to \mathbb{R}^K$, and our encoder function class is expressive enough to realize such a parameterization in its first $K$ components.*

*If an encoder realizing the manifold parameterization $e^*$ is optimal (together with a suitable optimal decoder), then the manifold is $K'$-parameterizable with $K' = K$, and*

$$L_{K'}[(e^*, d^*)] = 0.$$

The proof can be found in Appendix A.3.2.

**Definition 3.3 ($K'$-best-approximating encoder)**
*Suppose the data is supported on a $K$-dimensional manifold, which is $K'$-parameterizable. If a solution $(e^*, d^*)$ satisfying Eq. 12 is also optimal among all dimension-$K'$ encoders:*

$$(e^*, d^*) \in \underset{e,d}{\operatorname{argmin}} \sum_{k=0}^{K'} L_k[e, d],$$

*we denote it as the $K'$-best-approximating encoder (with an accompanying optimal decoder).*

Before stating the main result of this Section, we summarize how the pieces fit. Definition 3.1 is a notion *about the data manifold*: "$K'$-parameterizable" means that the energy score term $L_k[e, d]$ attains its global minimum at $k = K'$ for *some* encoder. Def. 3.3 is, conversely, a statement about the *model*: a $K'$-best-approximating encoder (i) achieves that globally minimal $K'$-term *and* (ii) is globally optimal among all $K'$-latent models. Theorem 3.4 below then describes the properties of such encoders, *assuming* they exist.

**Theorem 3.4 (Extraneous latents of a $K'$-best-approximating encoder are uninformative)**
*Suppose the data is supported on a $K$-dimensional manifold, which is $K'$-parameterizable. Then the $K'$-best-approximating encoder is also the optimal solution when optimizing across all $p$ dimensions, with the dimensions $(K' + 1, \cdots, p)$ obeying:*

$$P_{d^*, e^*_{1:k}(X)} = P_{d^*, e^*_{1:K'}(X)}, \quad \forall k \in [K' + 1, \ldots, p]. \tag{13}$$

*Furthermore, the dimensions $(K' + 1, \ldots, p)$ of the encoder will be* conditionally independent *of $X$, given the relevant components $(e^*_1, \cdots, e^*_{K'})$:*

$$\boxed{X \perp\!\!\!\perp e^*_{K'+i}(X) \mid e^*_{1:K'}(X), \quad \forall i \in [1, \ldots, p - K'].} \tag{14}$$

*In other words, they will carry* no additional information *about the data distribution:*

$$I\left(X; e^*_{K'+i}(X) \mid e^*_{1:K'}(X)\right) = 0, \quad \forall i \in [1, \ldots, p - K'],$$

*where $I(\cdot; \cdot)$ is the mutual information.*

The proof is located in Appendix A.3.3 and is based on the global optimality of the $K'$-best-approximating encoder and the fact that the random variables $e^*_{1:k}(X)$ form a filtration with increasing $k$.

The $K'$-best-approximating encoder simply repeats the $K'$-dimensional distributional approximation in the extraneous dimensions. The "extra" coordinates may be deterministic functions of the preceding ones (or the data) or stochastic, yet conditionally *independent* of $X$. This result naturally extends the property of PCA, which finds the principal linear subspace of the data, to a *nonlinear* encoding.

The following Corollary 3.5 is a special case for when an encoder exactly parameterizes a $K$-manifold and is globally optimal: it is a *sufficient* condition for Thm. 3.4, and reduces the existence question to the encoder being globally optimal.

**Corollary 3.5 (Optimal encoders that exactly parameterize the manifold output the data distribution)**
*Consider the case of an exactly parameterizable manifold with parameterization $e_{1:K}$. If this encoder is optimal among $K$-dimensional encoders, then it is a $K$-best-approximating encoder and Theorem 3.4 holds with $K' = K$. Furthermore, together with an accompanying optimal decoder, it will output exactly the data distribution using the first $K$ dimensions:*

$$d^* \left( e_{1:K}(X), \epsilon_{K+1:p} \right) \overset{a.s.}{=} X \sim P_{data}. \tag{15}$$

The proof is combined with that of Prop. 3.2 and can be found in Appendix A.3.2. The capacity of an autoencoder architecture to parameterize sufficiently "nice" manifolds was recently demonstrated in [5]; while the same remains as a future work for the DPA, it is certainly plausible.

Finally, the following remark addresses the attainability of the global optimality condition for the parameterizing encoder in Cor. 3.5:

**Remark 3.6**
*Consider the case of an exactly parameterizable $K$-dimensional manifold with parameterization $e_{1:K}$. Then, typically when $p \gg K$, such an encoder is an optimal encoder (for $K$ and $p$ dimensions), with the results 3.4 and 3.5 holding.*

A more precise statement is given in Appendix A.3.4; a short sketch is that since the parameterizing encoder achieves zero loss on the extraneous dimensions and cannot do "too bad" on the relevant ones, as $p \gg K$ this naturally leads to it being the globally optimal encoder.

# 4 Experiments

So far, the results presented are at the population level. In all the experiments below, the decoder is an *Engression* network, for which non-asymptotic finite-sample error bounds are available—see Thm. 3 in [19], so deviations from the population predictions shrink with the sample size. The encoder is a standard MLP throughout, so usual generalization arguments apply.

## 4.1 Level Set Score Alignment

We present score–alignment results for examples where the data density $P_{\text{data}}$ is known. In order of increasing complexity, we consider a standard multivariate Normal, a Gaussian mixture, and the Müller–Brown potential, which is a standard benchmark in molecular simulations.

The first two examples, shown in Fig. 1, train a DPA on 10,000 samples from either a standard Normal (*a*) or a three-component Gaussian mixture (*b*). Since the intrinsic dimension $K = 2 = p$, we visualize both encoder coordinates. Each panel shows a heatmap of the latent together with selected encoder level sets as contours. To accompany Fig. 1, Tab. 1 reports the *absolute cosine similarity* between the two *normal-space vectors* in Eq. 7 — the level-set vector and the data score after projection — evaluated on a $100 \times 100$ grid and restricted to points with density $> 0.5\%$ of the maximum (further details can be found in App. B.2). The alignment is essentially perfect in both datasets. For the standard normal (Fig. 1 *a*)), the level-set centers $c(X)$ should roughly coincide with the mean of the data distribution due to their geometry. Because the standard Gaussian is rotationally symmetric, DPA can choose any orthogonal pair of encoder directions without changing the loss. As the two components are orthogonal, we can examine each component's alignment with the score independently, as each should *approximately* satisfy Theorem 2.6. The first component thus roughly

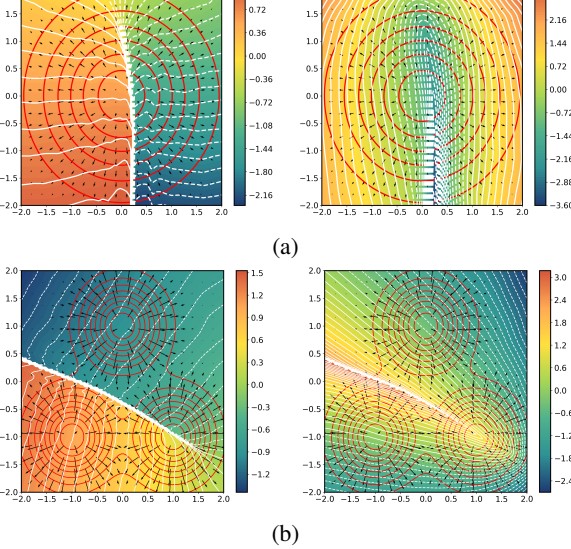

(a)

(b)

Figure 1: **Gaussian examples.** *a)* standard Normal; *b)* Gaussian mixture. Red contours: data density; black arrows: score. *Left*: first latent; *right*: second.

Table 1: Absolute cosine similarity between the normal-space vectors in Eq. 7 (points with density $> 0.5\%$ of its maximum kept).

| Dataset | Latent comp. | Mean abs. cosine | Std | 95%-ile | Points kept |
|---------|--------------|------------------|-----|---------|-------------|
| Standard Normal | 0 | 1.00 | 0.00 | 1.00 | 5 088 |
| Standard Normal | 1 | 1.00 | 0.00 | 1.00 | 5 088 |
| Gaussian Mixture | 0 | 1.00 | $3.1 \times 10^{-8}$ | 1.00 | 4 729 |
| Gaussian Mixture | 1 | 1.00 | $3.0 \times 10^{-8}$ | 1.00 | 4 729 |

encodes the polar angle and the second the radius, both parameterized with an increasing value of the latent $z$.

The first component attempts to *minimize* both sides of Eq. 7, that is, have both $y - c(X)$ and $\nabla_y \log P_{data}(y)$ lie in the tangent space of the level set. The second component, on the other hand, attempts to *maximize* both sides of Eq. 7 by having $y - c(X)$ and $\nabla_y \log P_{data}(y)$ almost entirely normal to the level set. Thus, we obtain (approximately) level sets that are orthogonal to the score gradient. The encoding also exhibits other desirable properties, such as all the level sets being connected and a smooth, directed variation of the latent.

For the Gaussian mixture with non-symmetric centers (Fig. 1 *b*), the polar picture persists but without rotational symmetry. The region of high density of the level sets coincides with the near-zero gradient region of the data distribution where the contributions of the three components cancel out. As the distribution is no longer symmetric, the score alignment must be understood *jointly* across latents rather than per-coordinate; nevertheless, the numeric alignment remains near-perfect.

The Müller–Brown potential [16] is a standard two-dimensional benchmark in computational chemistry. It features three minima and several saddle points; the minima are marked by red dots in Fig. 2, with potential contours overlaid in red. Physically, the minima represent metastable states, and the potential is designed to simulate chemical processes where a molecule undergoes (potentially rare) transitions between different configurations (states). The data distribution is the Boltzmann distribution for the Müller-Brown Potential $U(x)$ at a (known) temperature $T$:

$$P_{\text{data}}(x; T) = \frac{1}{Z} \exp\bigl(-U(x)/k_B T\bigr),$$

where $k_B$ is the Boltzmann constant. We generate samples by running Brownian dynamics (typically initialized at the minima). After re-arranging Eq. 7, we obtain for each level set defined by $X$:

$$\vec{F}(y)\, D_{e*}^\top = -\nabla_y\, U(y)\, D_{e*}^\top(y) = 2\, k_B T\, \frac{y - c(X)}{\frac{V(X)}{Z(X)} - \|y - c(X)\|^2}\, D_{e*}^\top(y), \qquad (16)$$

where $\vec{F}(y)$ is the force (field) at position $y$. This implies that the level set geometry is determined by the normal components of the force field and could, in principle, be recovered from the latter. For training DPA, we discard trajectory information and consider the data as i.i.d. samples from an unknown distribution. This contrasts with time-series approaches such as VAMPnets [15], which explicitly assume a Markovian structure of the process. Conversely, in existing i.i.d. autoencoder approaches, such as [7; 4], the authors commonly use an *iterative procedure* to find a good encoding: first, encode the data produced by an unbiased simulation (such as the data used here), then use the resulting encoding to add bias to the potential and run another simulation, then encode the data again, and so on until convergence. Both steps of this procedure are potentially computationally expensive for larger problems.

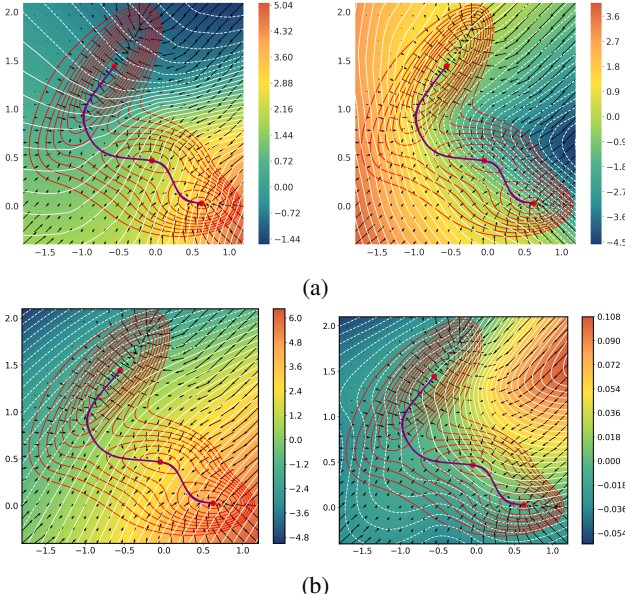

Figure 2: Müller–Brown potential: encoder level sets and comparisons. Red contours: potential; black arrows: potential gradient; purple: the MFEP. **(a)** First (left) and second (right) DPA components. **(b)** First component for an Autoencoder (left) and a VAE (right).

A quantity of considerable interest is the minimum free-energy path (MFEP), shown in purple in Fig. 2 and computed by the string method [8]. The MFEP connects minima and is everywhere tangent to $\nabla U$, representing the "least-energy-costly" transition between the former. In contrast to iterative pipelines, DPA approximates the MFEP from a *single* fit of unbiased data — even though samples between minima are very scarce (cf. App. B.1). By Eq. 16, the normal components of the encoder's level sets align with the force field; thus, the *first* DPA component provides a monotone (approximate) parameterization of the MFEP in its latent $z$. Along the inter-minimum corridor, its level sets are (approximately) *tangential* to $\nabla U$ and intersect the high-density regions orthogonally, placing $c(X)$ near the MFEP; following $\nabla_x e(x)$ from the top-left minimum with increasing $z$ thus traces the MFEP. For the *second* component, the level sets are approximately orthogonal to the gradient in the regions between the minima, and the level set $\{e(x) \approx 0\}$ approximately traces out the MFEP. In Fig. 2 (b), the Autoencoder captures the general direction of the transition but does not parameterize the MFEP, whereas the VAE fails to encode the transition pathway entirely.

We quantify the MFEP parameterization ability in Tab. 2 while extending the comparison to $\beta$-VAE [11] and $\beta$-TCVAE [6]. DPA achieves the best MFEP agreement across all distance metrics, and its first component is *always* the best parameterizing one ($\bar{z}_\parallel$), in line with the *principal-components* nature of the method (details and further results in App. B.3).

Table 2: Evaluation of distances between the closest parameterization and the true MFEP: each entry is the mean $\pm$ s.d. over the 24 seeds kept after discarding the worst-Chamfer run for every model. $\bar{d}_{\bullet \to \bullet}$ represent average closest distances for both directions, while 95th-perc. error is the larger of the two 95th-percentiles for the two closest distance sets. Lower numbers are better for every distance metric.

| Model | Param. comp. $\bar{z}_{\|}$ | Chamfer $\bar{d}_{\mathrm{C}}$ | Hausdorff $\bar{d}_{\mathrm{H}}$ | Mean 95th-perc. error | $\bar{d}_{\mathrm{MFEP}\to\mathrm{path}}$ | $\bar{d}_{\mathrm{path}\to\mathrm{MFEP}}$ |
|---|---|---|---|---|---|---|
| AE | $0.54 \pm 0.51$ | $0.387 \pm 0.113$ | $0.804 \pm 0.142$ | $0.760 \pm 0.110$ | $0.467 \pm 0.128$ | $0.306 \pm 0.100$ |
| DPA | $0.00 \pm 0.00$ | $\mathbf{0.262 \pm 0.053}$ | $\mathbf{0.730 \pm 0.317}$ | $\mathbf{0.567 \pm 0.212}$ | $\mathbf{0.289 \pm 0.066}$ | $0.236 \pm 0.047$ |
| VAE | $0.62 \pm 0.49$ | $0.515 \pm 0.469$ | $1.461 \pm 0.973$ | $1.311 \pm 0.980$ | $0.541 \pm 0.217$ | $0.490 \pm 0.809$ |
| $\beta$-VAE | $0.5 \pm 0.51$ | $0.450 \pm 0.288$ | $1.172 \pm 0.512$ | $1.051 \pm 0.477$ | $0.539 \pm 0.180$ | $0.360 \pm 0.462$ |
| $\beta$-TCVAE | $0.375 \pm 0.49$ | $0.377 \pm 0.077$ | $1.378 \pm 0.501$ | $1.228 \pm 0.433$ | $0.591 \pm 0.180$ | $\mathbf{0.164 \pm 0.101}$ |

Table 3: Determinism diagnostics for extraneous latents $U$. "ID-drop" reports the $2.5\%$, $50\%$, $97.5\%$ bootstrap quantiles (200 resamples). Dataset details in App. B.4.

| Dataset | $\mathbf{R^2}$ | ID-drop | $\mathbf{H(U\,|\,Z)}$ [nats] |
|---|---|---|---|
| Gaussian line | 0.9997 | 0.0112 / **0.0122** / 0.0132 | $-7.259$ |
| Parabola | 0.9997 | 0.0046 / **0.0048** / 0.0051 | $-9.190$ |
| Exponential | 0.9996 | 0.0045 / **0.0047** / 0.0050 | $-9.162$ |
| Helix slice | 0.9995 | 0.0041 / **0.0043** / 0.0045 | $-6.090$ |
| Grid sum | 0.9986 | 0.0023 / **0.0029** / 0.0034 | $-2.759$ |
| S-curve | 0.9996 | $-0.0031$ / $-0.0014$ / 0.0000 | $-1.762$ |
| S-curve ($\beta = \mathbf{2}$) | 0.9990 | 0.0030 / **0.0034** / 0.0037 | $-2.600$ |
| Swiss-roll (3D) | 0.9872 | $-0.0048$ / $-0.0025$ / $-0.0003$ | $-0.042$ |

## 4.2 Independence of Extraneous Latents

Theorem 3.4 states that, at the optimum, latent coordinates beyond the manifold are uninformative given the informative block $Z$. This can be satisfied by two regimes: (i) on many manifolds, the "extra" latents $U$ become (nearly) deterministic functions of $Z$, and (ii) they are stochastic, yet conditionally independent of the data $X$ given $Z$.

**Deterministic regime.** We test whether $U \approx g(Z)$ using three diagnostics (details in App. B.4):
(i) $R^2$ of a nonlinear regressor of $U$ on $Z$ (near-1 indicates determinism),
(ii) the intrinsic-dimension *drop* when using $Z$ vs. $(Z, U)$ via the Levina–Bickel MLE: drop $\approx 0$ implies no additional degrees of freedom in $U$, and
(iii) an estimate of the conditional entropy $H(U \mid Z)$: very negative values are consistent with near-determinism. Across datasets, we observe $R^2 \approx 1$, near-zero ID-drop, and strongly negative $H(U \mid Z)$; see Table 3. Results include a $\beta = 2$ row, which still exhibits uninformative extraneous latents (despite the theoretical requirement $\beta \in (0, 2)$).

**Stochastic but uninformative regime.** In this case, we test for the null hypothesis $U \perp\!\!\!\perp X \mid Z$ directly. For the Gaussian line dataset, a *conditional randomization test* yields p-values consistent with the null hypothesis distribution – Uniform$[0, 1]$; a Kolmogorov–Smirnov test gives $D = 0.061$, a p-value $p = 0.822$ with $4\%$ of 100 replications below 0.05, confirming conditional independence (details and further results in App. B.4).

## 5 Related work

For denoising (and related contractive) autoencoders, Alain and Bengio [1] derive a formula showing that, for each *fixed data point*, *asymptotically* as the noise approaches zero, the difference between the reconstructed data vector and the original will tend to the score of the data. Using our notation:

$$d^*(e^*_\sigma(x)) = x + \sigma^2 \nabla_x \log P_{data}(x) + o\left(\sigma^2\right) \quad \text{as } \sigma \to 0, \forall x \text{ fixed}. \tag{17}$$

In Bengio et al. [3], the authors further relate the score to the first two *local moments* of the data distribution (moments restricted to a $\delta$-ball around a given point) and propose asymptotically valid

estimators as *both* the noise and $\delta \to 0$. In contrast, the first two moments appear directly in Eq. 7 as *level-set* moments, yielding a global geometric constraint that holds *almost surely*. It is also interesting to note that the results presented here are also not connected to any sort of regularization (whether denoising, contractive, or other), as they arise directly from the distributional regression using the energy score.

A concurrent line of work underscores the advantage of energy-score training over (denoising) score matching. Shen et al. [20] replace diffusion's score-matching objective across noise levels [21] with a sequence of energy-score regressions (as in the DPA), achieving comparable distributional fidelity in far fewer steps. This mirrors the contrast between the asymptotic nature of Alain and Bengio [1] (linked to diffusion via Vincent [22]) and our exact, non-asymptotic identities. Mirroring the (implicit) recovery of the Müller-Brown force field presented here, the score matching property of diffusion models has recently been used in [2] to obtain force fields.

On the informativeness of extraneous dimensions, Liu et al. [14] analyze Chart Autoencoders [17] that build on denoising autoencoders and show that, for both exactly and approximately parameterizable manifolds, the optimal reconstruction error decays exponentially in the manifold's intrinsic (not ambient) dimension, provided the corruption acts normal to the manifold — thus requiring geometric knowledge which DPA does not. For VAEs with learnable decoder variance, Zheng et al. [23] prove that, on simple Riemannian manifolds, the optimal decoder variance collapses to zero and the VAE loss scales with $(p - k) \log \sigma^2$, making the intrinsic dimension identifiable. Prior work on disentangled representation learning has also investigated the phenomenon of extraneous dimensions (depending on the formulation) being "'turned-off", typically utilizing extra regularization terms in the loss [11; 12]. In contrast, for the DPA we prove that extra latents are *exactly* (i.e., not limiting-behavior) uninformative — across both manifold scenarios — without manifold knowledge, additional regularization, or specialized architectures.

## 6 Discussion

This paper establishes two exact properties of the Distributional Principal Autoencoder (DPA). *First*: a closed-form relation between the optimal level sets of the encoder and the score of the data distribution. *Second*: if the data lie on a manifold that can be approximated by the encoder, the encoder's components beyond the dimension of the manifold (or its best approximation) are conditionally independent of the data and therefore carry no additional information.

The first identity is global and non-asymptotic, significantly strengthening related results for denoising/contractive autoencoders [1]. Likewise, the second main result unifies and expands on results from manifold learning and disentangled representation learning literature [14; 23]. Crucially, they hold *simultaneously*: a single DPA fit can be successful in both reconstruction/data-distribution learning through score alignment, *and* disentanglement/intrinsic dimension determination via conditional independence. This contrasts with almost all unsupervised learning methods where the two properties are a trade-off: e.g., this trade-off is *precisely* what the titular $\beta$ in $\beta$-VAEs [11] controls. Taken together, the two properties also yield a PCA-like picture: a nested ordering of coordinates and a rigorous cutoff via conditional independence, with the "principal" directions determined by the data-distribution geometry.

Our statements assume sufficient "niceness" of the data distribution (allowing, e.g. for full-rank Jacobians on enough level sets for the Theorem to be relevant) and sufficient encoder expressiveness to approximate the manifolds. While we derive expressions for general $\beta$, the most explicit level-set–score identity uses $\beta = 2$, where the optimal decoder need not be unique; empirically we did not observe degeneracy. All results are at the population optimum; finite samples and optimization errors can introduce deviations, as discussed with each result.

As demonstrated, score alignment in practice enables (approximate) recovery of force fields from samples drawn from the Boltzmann distribution, yielding a single-fit approximation to minimum free-energy paths and suggesting methods that could significantly speed up molecular simulations. It also suggests recovering densities from learned encoder level sets and motivates energy-score–based generative modeling. On the representation side, the conditional independence of extraneous latents also presents a concrete approach to intrinsic dimension determination beyond heuristics such as scree plots: one can directly test for conditional independence in the learned encoding.

## Acknowledgments

We wish to thank Nicolai Meinshausen and Felipe Maia Polo for helpful discussions.

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

# A  Theoretical Appendix

## A.1  Notation summary

Table 4: Notation used throughout the paper.

| Symbol | Meaning |
| --- | --- |
| $P_{\text{data}}$ | Data-generating density |
| $X \in \mathbb{R}^p$ | Data vector drawn from $P_{\text{data}}$ |
| $e : \mathbb{R}^p \to \mathbb{R}^k$ | Encoder |
| $d : \mathbb{R}^k \times \mathbb{R}^{p-k} \to \mathbb{R}^p$ | Stochastic decoder |
| $P_{e,x}^*$ | Oracle reconstructed distribution (ORD) conditioned on $e(X) = e(x)$ |
| $Z(X)$ | Level–set mass at $X$ |
| $c(X)$ | Level–set center-of-mass at $X$ |
| $V(X)$ | Level–set variance at $X$ |
| $D_e(y)$ | Jacobian of the encoder at point $y$ |

## A.2  Proofs of Section 2

First, let's refresh the assumptions used in this section:

**Global assumptions** 2.3: $P_{\text{data}} \in C^1$, integrable; $e \in C^1$, Lipschitz.

**Local rank** 2.4 rank $D_e(y) = k$ a.e. on the level set $\mathcal{L}_e(X)$ for the $X$ under consideration.

The local rank assumption will be invoked when needed, while the global assumptions can be taken to hold throughout.

### A.2.1  Proof of Lemma 2.5

Writing out the first variation condition, we have:

$$\left. \frac{\delta}{\delta e} \right|_{e^*} \mathbb{E}_{X \sim P_{data}} \left[ \mathbb{E}_{Y,Y' \overset{\text{iid}}{\sim} P_{e,X}^*} \left[ \|Y - Y'\|^\beta \right] \right] = 0. \tag{18}$$

Equivalently, $\forall \eta$, we have for $Y \sim P_{e+\varepsilon\eta,X}^*$:

$$(e^* + \varepsilon\eta)(Y) \overset{d}{=} (e^* + \varepsilon\eta)(X), \tag{19}$$

by the definition of the ORD. We adopt the same assumptions on $\eta$ as we did on $e$ as $e + \varepsilon\eta$ needs to be a valid encoder. Furthermore, we will be dropping the $*$ in $e^*$ — we will be assuming $e$ to be optimal in the arguments to follow to clear up the notation.

Now, as is the norm in calculus of variations, assume the following form of the variation for small $\varepsilon$:

$$e(Y) + \varepsilon\eta(Y) \overset{d}{\approx} e(X) + \varepsilon\eta(X), \tag{20}$$

for $Y \sim P_{e+\varepsilon\eta,X}^*$, i.e. $\{y : (e + \varepsilon\eta)(y) = (e + \varepsilon\eta)(X)\} = L_{(e^*+\varepsilon\eta)(X)}$, that is, from the *perturbed* level set.

More precisely, to the first order in $\varepsilon$, we have:

$$e(Y) + \varepsilon\eta(Y) = e(X) + \varepsilon\eta(X) + \mathcal{O}(\varepsilon^2)$$

Which leads to the following property of the perturbation $\eta$:

$$e(Y) - e(X) = -\varepsilon\big(\eta(Y) - \eta(X)\big) + \mathcal{O}(\varepsilon^2). \tag{21}$$

for $Y \sim P^*_{e+\varepsilon\eta,X}$.

Furthermore, due to $e, \eta \in \mathcal{C}^1$ and both being Lipschitz, we have $\eta(Y) - \eta(X) = \mathcal{O}(1)$.

$\forall \eta$, the stationarity condition 18 thus becomes:

$$\frac{d}{d\varepsilon}\bigg|_{\varepsilon=0} \mathbb{E}_{X \sim P_{data}} \left[ \mathbb{E}_{Y,Y' \overset{\mathrm{iid}}{\sim} P^*_{e+\varepsilon\eta,X}} \left[ \|Y - Y'\|^\beta \right] \right] = 0 \tag{22}$$

As

Using the Dirac delta functions, we can express the ORD as:

$$P^*_{e,X}(y) = \frac{P_{data}(y)\,\delta(e(y) - e(X))}{\int P_{data}(z)\,\delta(e(z) - e(X))\,dz}, \tag{23}$$

and, equivalently, the "perturbed" ORD as:

$$P^*_{e+\varepsilon\eta,X}(y) = \frac{P_{data}(y)\,\delta((e + \varepsilon\eta)(y) - (e + \varepsilon\eta)(X))}{\int P_{data}(z)\,\delta((e + \varepsilon\eta)(z) - (e + \varepsilon\eta)(X))\,dz} \tag{24}$$

Using the definition of the *level-set mass* 5:

$$Z(X) = \int P_{data}(z)\,\delta(e(z) - e(X))\,dz,$$

we can define the "perturbed" mass:

$$Z_\varepsilon(X) = \int P_{data}(z)\,\delta((e + \varepsilon\eta)(z) - (e + \varepsilon\eta)(X))\,dz$$

Writing out Eq. 22 explicitly:

$$0 = \mathbb{E}_{X \sim P_{data}} \left[ \iint \|y - y'\|^\beta \frac{d}{d\varepsilon}\bigg|_{\varepsilon=0} \left[ P^*_{e+\varepsilon\eta,X}(y)\, P^*_{e+\varepsilon\eta,X}(y') \right] dy\, dy' \right],$$

where we used Leibnitz' rule to move the derivative inside the integral, which is justified since the ORD terms $P^*_{e+\varepsilon\eta,X}(y)$ vanish exponentially (as they inherit the tail behavior from $P_{data}$, while $\|y - y'\|$ grows at most polynomially, hence a dominated-convergence requirement is satisfied.

We note:

$$\frac{d}{d\varepsilon}\bigg|_{\varepsilon=0} P^*_{e+\varepsilon\eta,X}(y) = P^*_{e,X}(y)\,\frac{d}{d\varepsilon}\bigg|_{\varepsilon=0} \log(P^*_{e+\varepsilon\eta,X}(y)). \tag{25}$$

Using the product rule on the product of the two ORDs:

$$\frac{d}{d\varepsilon}\bigg|_{\varepsilon=0} \left[ P^*_{e+\varepsilon\eta,X}(y)\, P^*_{e+\varepsilon\eta,X}(y') \right]$$
$$= \frac{d}{d\varepsilon}\bigg|_{\varepsilon=0} P^*_{e+\varepsilon\eta,X}(y) \cdot P^*_{e,X}(y') + P^*_{e,X}(y) \cdot \frac{d}{d\varepsilon}\bigg|_{\varepsilon=0} P^*_{e+\varepsilon\eta,X}(y')$$

and substituting the above, we obtain

$$\frac{d}{d\varepsilon}\bigg|_{\varepsilon=0} \left[ P^*_{e+\varepsilon\eta,X}(y)\, P^*_{e+\varepsilon\eta,X}(y') \right] = P^*_{e,X}(y)P^*_{e,X}(y') \left[ \frac{d}{d\varepsilon}\bigg|_{\varepsilon=0} \log(P^*_{e+\varepsilon\eta,X}(y)) + \frac{d}{d\varepsilon}\bigg|_{\varepsilon=0} \log(P^*_{e+\varepsilon\eta,X}(y')) \right]$$

Thus, Eq. 22 is equivalent to:

$$0 = \mathbb{E}_{X \sim P_{data}} \left[ \iint \|y - y'\|^\beta P^*_{e,X}(y)P^*_{e,X}(y') \left[ \frac{d}{d\varepsilon}\bigg|_{\varepsilon=0} \log(P^*_{e+\varepsilon\eta,X}(y)) + \frac{d}{d\varepsilon}\bigg|_{\varepsilon=0} \log(P^*_{e+\varepsilon\eta,X}(y')) \right] dy\, dy' \right] \tag{26}$$

Now, let's examine the derivatives of the *perturbed* log-ORDs explicitly using 24 and the linear variation approximation 20:

$$\log(P^*_{e+\varepsilon\eta,X}(y)) = \log(P_{data}(y)) + \log(\delta(e(y) + \varepsilon\eta(y) - e(X) - \varepsilon\eta(X)))$$
$$- \log\left(\int P_{data}(z)\,\delta(e(z) + \varepsilon\eta(z) - e(X) - \varepsilon\eta(X))dz\right)$$

After taking the derivative evaluated at $\varepsilon = 0$, we get for the second term:

$$(\eta(y) - \eta(X))\frac{\delta'(e(y) - e(X))}{\delta(e(y) - e(X))}$$

and equivalently for the second log-ORD for $y'$.

We have used $\frac{d}{d\varepsilon}\big|_{\varepsilon=0}\delta(f(x) + \varepsilon g(x)) = g(x)\,\delta'(f(x))$, where we leave the derivative of the Dirac delta undefined, for now.

For the third term we get:

$$\frac{d}{d\varepsilon}\Big|_{\varepsilon=0} - \log Z_\varepsilon(X) = \frac{-1}{Z(X)}\int P_{data}(z)\frac{d}{d\varepsilon}\Big|_{\varepsilon=0}\delta(e(z) + \varepsilon\eta(z) - e(X) - \varepsilon\eta(X))\,dz$$
$$= \frac{-1}{Z(X)}\int P_{data}(z)\,(\eta(z) - \eta(X))\,\delta'(e(z) - e(X))\,dz,$$

where we used our definition of the level-set mass 5.

The first-order variation condition Eq. 22 thus becomes:

$$0 = \mathbb{E}_{X \sim P_{data}}\left[\iint \|y - y'\|^\beta \frac{P_{data}(y)P_{data}(y')}{Z(X)^2}\,\delta(e(y) - e(X))\,\delta(e(y') - e(X))\right.$$
$$\cdot\left\{(\eta(y) - \eta(X))\frac{\delta'(e(y) - e(X))}{\delta(e(y) - e(X))} + (\eta(y') - \eta(X))\frac{\delta'(e(y') - e(X))}{\delta(e(y') - e(X))}\right.$$
$$\left.\left. - \frac{2}{Z(X)}\int P_{data}(z)\,(\eta(z) - \eta(X))\,\delta'(e(z) - e(X))\,dz\right\}dy\,dy'\right] \tag{27}$$

We want to use integration by parts to get rid of the pesky derivatives of the Dirac delta functions.

Now $\delta(e(z) - e(X))$ maps $\mathbb{R}^k \to \mathbb{R}$, and $e\colon \mathbb{R}^p \to \mathbb{R}^k$. By the higher-dimensional chain rule:

$$D\left(\delta(e(z) - e(X))\right) = D_{\delta(u)}\big|_{u=e(z)-e(X)}\,D_{e(z)-e(X)}(z) = \nabla_u\delta(u)\big|_{u=e(z)-e(X)}\,D_e(z),$$

where $D$ is the total derivative, and $D_\bullet$ the Jacobian. We are treating the gradient $\nabla_u\delta(u)$ as a row-vector to preserve the overall chain rule form.

Above, we have used $\delta'(e(z) - e(X))$ as a shorthand for the $k$-dimensional gradient $\nabla_u\delta(u)$ evaluated at $u = e(z) - e(X)$, and the product of the delta function with the other terms is a dot product from the higher-dimensional chain rule:

$$\frac{d}{d\varepsilon}\delta(\mathbf{u} = (e(y) - e(X)) + \varepsilon(\eta(y) - \eta(X)))\Big|_{\varepsilon=0} = \nabla_{\mathbf{u}}\delta(\mathbf{u})\big|_{\mathbf{u}=e(z)-e(X)} \cdot [\eta(z) - \eta(X)].$$

Thus, we have terms of the following form:

$$-\frac{1}{Z(X)}\int P_{data}(z)[\eta(z) - \eta(X)] \cdot \nabla_{\mathbf{u}}\delta(\mathbf{u})\Big|_{\mathbf{u}=e(z)-e(X)}\,dz$$

Next, the gradients of the $\delta$ function are to be interpreted distributionally: for any (smooth, compactly supported) test function $\psi(z)\colon \mathbb{R}^p \to \mathbb{R}$ that vanishes at the boundary, we have:

$$\int \psi(z)\nabla_z\delta(e(z) - e(X))\,dz = -\int \delta(e(z) - e(X))\nabla_z\psi(z)\,dz$$

or, for the dot product with a vector-valued function $\varphi$ that, likewise vanishes:

$$\int \varphi(z) \cdot \nabla_z \delta(\ldots) \, dz = -\int \nabla_z \cdot \varphi(z) \, \delta(\ldots) \, dz$$

by a "distributional" integration by parts (via the divergence theorem).

Thus, the above chain rule is to be interpreted as the following distributional equality:

$$\int \psi(z) \nabla_z \delta(e(z) - e(X)) \, dz = \int \psi(z) \left[\nabla_{\mathbf{u}} \delta(\mathbf{u})\right]\Big|_{\mathbf{u}=e(z)-e(X)} D_e(z) \, dz. \qquad (28)$$

Now, let's apply this to Eq. 27, starting with the *third term* inside the square brackets:

$$\int P_{data}(z) \, (\eta(z) - \eta(X)) \, \delta'(e(z) - e(X)) \, dz$$

Denote $\varphi(z) = P_{data}(z) \, (\eta(z) - \eta(X)) \in \mathbb{R}^k$.

Introduce a scalar test function $f$, i.e., examine the following integral:

$$\int \varphi(z) \cdot \left[\nabla_{\mathbf{u}} \delta(\mathbf{u})\right]\big|_{u=e(z)-c} f(z) \, dz$$

Now define: $G(z) := \varphi(z) f(z) \in \mathbb{R}^k$. Note that $G$ vanishes at the boundary due to the $P_{data}$ factor, hence we don't need that requirement on $f$. Thus, using $u = e(z) - e(X)$:

$$\int \varphi(z) \cdot \left[\nabla_{\mathbf{u}} \delta(\mathbf{u})\right] f(z) \, dz = \int G(z) \cdot \left[\nabla_{\mathbf{u}} \delta(\mathbf{u})\right] \, dz$$

Now examine the integral where we multiply the integrand with $D_e^\top(z)$ from the left, and $D_e(z)$ from the right:

$$\int \left(D_e^\top(z) \, G(z)\right) \cdot \left(\left[\nabla_{\mathbf{u}} \delta(\mathbf{u})\right] D_e(z)\right) \, dz,$$

*assuming* the Jacobian $D_e(z)$ has *full row rank $k$* a.e. on the level set $L_{e(X)}$, as that is the integration domain induced by the $\delta$ function. In other words, this is the point at which we invoke Assumption 2.4. Note that, as stated in the discussion around this assumption, Lemma 2.5 and Theorem 2.6 are simply *silent* on level sets that do not satisfy this condition. [2]

We will not be simplifying the linear algebra, but instead note that by the distributional equality of the chain rule — Eq. 28, this must equal:

$$\int \left[D_e^\top(z) \, G(z)\right] \cdot \left(\left[\nabla_{\mathbf{u}} \delta(\mathbf{u})\right] D_e(z)\right) \, dz = \int \left(D_e^\top(z) \, G(z)\right) \cdot \left[\nabla_z \delta(e(z) - e(X))\right] \, dz$$

Since this holds for any $f$, we get for $f = 1$:

$$\int \varphi(z) \cdot \left[\nabla_{\mathbf{u}} \delta(\mathbf{u})\right] dz = \int \left(D_e^\top(z) \, \varphi(z)\right) \cdot \left[\nabla_z \delta(e(z) - e(X))\right] dz \qquad (29)$$

We can now integrate by parts and obtain:

$$\int \left(D_e^\top(z) \, \varphi(z)\right) \cdot \left[\nabla_z \delta(e(z) - e(X))\right] dz = -\int \nabla_z \cdot \left(D_e^\top(z) \, \varphi(z)\right) \delta(e(z) - e(X)) \, dz \quad (30)$$

We have discarded the boundary terms since $\varphi \to 0$ as $\|z\| \to \infty$ due to $P_{data}$ being a valid (integrable) density, and $\eta$ being Lipschitz.

Thus, we arrive at the following expression for the third term:

$$\int P_{data}(z)(\eta(z) - \eta(X)) \, \delta'(e(z) - e(X)) \, dz = -\int \nabla_z \cdot [D_e^\top(z) \, P_{data}(z) \, (\eta(z) - \eta(X))] \, \delta(e(z) - e(X)) \, dz$$

---

[2] We believe that adopting a stricter requirement on the encoder's smoothness and continuity would allow us to invoke Sard's theorem and drop this assumption; however, we consider that scenario less realistic.

Let's now apply the same to the other two, "symmetric" terms in Eq. 27: Focus on the first one which depends on $y$. Ignoring the outer expectation over $X$, we have:

$$\iint \|y - y'\|^\beta \frac{P_{\text{data}}(y)P_{\text{data}}(y')}{Z(X)^2} \delta(e(y) - e(X))\, \delta(e(y') - e(X)) \left[ (\eta(y) - \eta(X)) \frac{\delta'(e(y) - e(X))}{\delta(e(y) - e(X))} \right] dy\, dy'$$

$$= \frac{1}{Z(X)^2} \int P_{\text{data}}(y')\, \delta(e(y') - e(X))\, dy' \int \|y - y'\|^\beta P_{\text{data}}(y)(\eta(y) - \eta(X))\, \delta'(e(y) - e(X))\, dy.$$

Note that the potentially problematic ratio $\frac{\delta'}{\delta}$ was merely to clean up the notation.

Denoting now $\varphi(y) = \|y - y'\|^\beta P_{\text{data}}(y)\,(\eta(y) - \eta(X))$, we get by the same integration by parts (cf. Eq. 30):

$$= \frac{-1}{Z(X)^2} \int P_{\text{data}}(y')\, \delta(e(y') - e(X))\, dy' \int \nabla_y \cdot [D_e^\top(y)\, \|y - y'\|^\beta P_{\text{data}}(y)(\eta(y) - \eta(X))]\, \delta(e(y) - e(X))\, dy$$

And, due to symmetry, for the other term:

$$= \frac{-1}{Z(X)^2} \int P_{\text{data}}(y)\, \delta(e(y) - e(X))\, dy \int \nabla_{y'} \cdot [D_e^\top(y')\, \|y - y'\|^\beta P_{\text{data}}(y')(\eta(y') - \eta(X))]\, \delta(e(y') - e(X))\, dy'$$

Putting it all together, we have the following equivalent of Eq. 27:

$$0 = \mathbb{E}_{X \sim P_{data}} \left[ \frac{1}{Z(X)^2} \int \int dy\, dy'\, \delta(e(y) - e(X))\, \delta(e(y') - e(X)) \right.$$

$$\left\{ - P_{\text{data}}(y')\, \nabla_y \cdot [D_e^\top(y)\, \|y - y'\|^\beta P_{\text{data}}(y)(\eta(y) - \eta(X))] \right.$$

$$- P_{\text{data}}(y)\nabla_{y'} \cdot [D_e^\top(y')\, \|y - y'\|^\beta P_{\text{data}}(y')(\eta(y') - \eta(X))]$$

$$\left. \left. + \frac{2}{Z(X)} \|y - y'\|^\beta P_{data}(y)P_{data}(y') \int \nabla_z \cdot [D_e^\top(z)\, P_{data}(z)\,(\eta(z) - \eta(X))]\, \delta(e(z) - e(X))\, dz \right\} \right]$$

$$(31)$$

We can now spot that the first two terms inside the curly brackets are identical since $Y, Y' \overset{\text{iid}}{\sim} P_{e,X}^*$, hence we can simplify. Thus, $\forall \eta$:

$$0 = \mathbb{E}_{X \sim P_{data}} \left[ \frac{2}{Z(X)^2} \int P_{\text{data}}(y')\, \delta(e(y') - e(X))\, dy' \int dy\, \delta(e(y) - e(X)) \right.$$

$$\left[ - \nabla_y \cdot [D_e^\top(y)\, \|y - y'\|^\beta P_{\text{data}}(y)(\eta(y) - \eta(X))] \right.$$

$$\left. \left. + \frac{1}{Z(X)} \|y - y'\|^\beta P_{data}(y) \int \nabla_z \cdot [D_e^\top(z)\, P_{data}(z)\,(\eta(z) - \eta(X))]\, \delta(e(z) - e(X))\, dz \right] \right]$$

$$(32)$$

The optimality condition will be zero in full generality — $\forall \eta$ — if the terms inside the brackets are zero, implying $\forall X$ *almost surely*:

$$\int P_{\text{data}}(y')\, \delta(e(y') - e(X))\, dy' \int \nabla_y \cdot [D_e^\top(y)\, \|y - y'\|^\beta P_{\text{data}}(y)(\eta(y) - \eta(X))]\, \delta(e(y) - e(X))\, dy$$

$$= \frac{1}{Z(X)} \int P_{\text{data}}(y')\, \delta(e(y') - e(X))\, dy' \int \|y - y'\|^\beta P_{data}(y)\, \delta(e(y) - e(X))\, dy$$

$$\int \nabla_z \cdot [D_e^\top(z)\, P_{data}(z)\,(\eta(z) - \eta(X))]\, \delta(e(z) - e(X))\, dz \tag{33}$$

This is the general balance equation that must hold on the level sets of the encoder (as these are induced by the $\delta$-functions in the integral).

Defining:

$$J_\beta\left(y'; X, \eta\right) = \int \nabla_y \cdot \left[D_e^\top(y) \left\|y - y'\right\|^\beta P_{\text{data}}(y)(\eta(y) - \eta(X))\right] \delta(e(y) - e(X)) \, dy,$$

$$S(X, \eta) = \int \nabla_z \cdot \left[D_e^\top(z) P_{data}(z)(\eta(z) - \eta(X))\right] \delta(e(z) - e(X)) \, dz,$$

and WLOG switching around $y$ and $y'$, we arrive at the desired expression:

$$\mathbb{E}_{Y \sim P_{e^*,X}^*} \left[J_\beta\left(Y; X, \eta\right)\right] = \frac{S(X, \eta)}{Z(X)} \, \mathbb{E}_{Y, Y' \overset{\text{iid}}{\sim} P_{e^*,X}^*} \left[\left\|Y - Y'\right\|^\beta\right]$$

$\square$

### A.2.2 Proof of Theorem 2.6

While Lemma 2.5 holds for any $\beta$, we now focus on the case of the squared Euclidean norm: $\beta = 2$. As we're continuing from the proof of the Lemma, we adopt the same assumptions and notation as above.

First, let's restate the definition of the *level-set center of mass* (Eq. 8):

$$c(X) = \frac{1}{Z(X)} \int y \, P_{data}(y) \, \delta(e(y) - e(X)) \, dy$$

Likewise, for the *level-set variance* (Eq. 9).

$$V(X) = \int \|y - c(X)\|^2 P_{data}(y) \, \delta(e(y) - e(X)) \, dy$$

Expanding the norm, we get:

$$\|y - y'\|^2 = \|y - c(X) + c(X) - y'\|^2 = \underbrace{\|y - c(X)\|^2}_{A} + \underbrace{\|y' - c(X)\|^2}_{B} \underbrace{-2\langle y - c(X), y' - c(X)\rangle}_{C}$$

On both sides of Eq. 33, the C term vanishes: First, the RHS:

$$\frac{-2}{Z(X)} \int P_{\text{data}}\left(y'\right) \delta(e(y') - e(X)) \, dy' \int \langle y - c(X), y' - c(X)\rangle \, P_{data}(y) \, \delta(e(y) - e(X)) \, dy \int \dots dz$$

$$= \frac{-2}{Z(X)} \int \langle y - c(X), \smallint(y' - c(X))P_{\text{data}}\left(y'\right) \delta(e(y') - e(X)) \, dy'\rangle \, P_{data}(y) \, \delta(e(y) - e(X)) \, dy \int \dots dz$$

$$= 0$$

where we used the bilinearity property of inner product (and Fubini's theorem), and:

$$\int (y' - c(X))P_{\text{data}}\left(y'\right) \delta(e(y') - e(X)) \, dy' = Z(X)c(X) - c(X)Z(X) = 0$$

by the definition of the center-of mass.

On the LHS, we get a term like:

$$\int P_{\text{data}}\left(y'\right) \delta(e(y') - e(X)) \, dy' \int \delta(e(y) - e(X))\nabla_y \cdot [\langle y - c(X), y' - c(X)\rangle P_{\text{data}}(y) \, D_e^\top(y)(\eta(y) - \eta(X))] \, dy$$

Since the divergence operator is again linear and acts on $y$-only, we can bring the integral over $y'$ inside the inner product, getting this term to vanish, again.

The B term gives us on the RHS:

$$\frac{1}{Z(X)} \int \|y' - c(X)\|^2 P_{\text{data}}\left(y'\right) \delta(e(y') - e(X)) \, dy' \int \delta(e(y) - e(X))P_{data}(y) \, dy$$

$$\int \nabla_z \cdot [D_e^\top(z) \, P_{data}(z) \, (\eta(z) - \eta(X))] \, \delta(e(z) - e(X)) \, dz = \frac{1}{Z(X)}V(X)Z(X) \int \dots dz$$

$$= V(X) \int \nabla_z \cdot [D_e^\top(z) \, P_{data}(z) \, (\eta(z) - \eta(X))] \, \delta(e(z) - e(X)) \, dz$$

On the LHS:

$$\int P_{\text{data}}(y')\,\delta(e(y') - e(X))\,dy' \int \delta(e(y) - e(X))\nabla_y \cdot [D_e^\top(y)\,\|y' - c(X)\|^2\,P_{\text{data}}(y)(\eta(y) - \eta(X))]\,dy$$

$$= V(X)\int \nabla_y \cdot [D_e^\top(y)P_{\text{data}}(y)(\eta(y) - \eta(X))]\,\delta(e(y) - e(X))\,dy$$

Thus, we have for the optimality condition 33 in terms of $c(X)$ and $V(X)$ on the RHS:

$$\frac{1}{Z(X)}\int P_{\text{data}}(y')\,\delta(e(y') - e(X))\,dy' \int \delta(e(y) - e(X))\|y - y'\|^2 P_{data}(y)\,dy$$

$$\int \nabla_z \cdot [D_e^\top(z)\,P_{data}(z)\,(\eta(z) - \eta(X))]\,\delta(e(z) - e(X))\,dz$$

$$= \frac{1}{Z(X)}\int P_{\text{data}}(y')\,\delta(e(y') - e(X))\,dy' \int \|y - c(X)\|^2 P_{data}(y)\,\delta(e(y) - e(X))\,dy \int \ldots dz$$

$$+ V(X)\int \nabla_z \cdot [D_e^\top(z)\,P_{data}(z)\,(\eta(z) - \eta(X))]\,\delta(e(z) - e(X))\,dz$$

$$= 2\,V(X)\int \nabla_z \cdot [D_e^\top(z)\,P_{data}(z)\,(\eta(z) - \eta(X))]\,\delta(e(z) - e(X))\,dz$$

On the LHS:

$$\int P_{\text{data}}(y')\,\delta(e(y') - e(X))\,dy' \int \nabla_y \cdot [D_e^\top(y)\,\|y - y'\|^2\,P_{\text{data}}(y)(\eta(y) - \eta(X))]\,\delta(e(y) - e(X))\,dy$$

$$= \int P_{\text{data}}(y')\,\delta(e(y') - e(X))\,dy' \int \delta(e(y) - e(X))\nabla_y \cdot [\|y - c(X)\|^2\,P_{\text{data}}(y)D_e^\top(y)(\eta(y) - \eta(X))]\,dy$$

$$+ V(X)\int \nabla_y \cdot [D_e^\top(y)P_{\text{data}}(y)(\eta(y) - \eta(X))]\,\delta(e(y) - e(X))\,dy$$

$$= Z(X)\int \nabla_y \cdot [\|y - c(X)\|^2\,P_{\text{data}}(y)D_e^\top(y)(\eta(y) - \eta(X))]\,\delta(e(y) - e(X))\,dy$$

$$+ V(X)\int \nabla_y \cdot [P_{\text{data}}(y)D_e^\top(y)(\eta(y) - \eta(X))]\,\delta(e(y) - e(X))\,dy$$

Thus:

$$Z(X)\int \nabla_y \cdot [\|y - c(X)\|^2\,P_{\text{data}}(y)D_e^\top(y)(\eta(y) - \eta(X))]\,\delta(e(y) - e(X))\,dy$$

$$+ V(X)\int \nabla_y \cdot [P_{\text{data}}(y)D_e^\top(y)(\eta(y) - \eta(X))]\,\delta(e(y) - e(X))\,dy$$

$$= 2\,V(X)\int \nabla_z \cdot [P_{data}(z)\,D_e^\top(z)(\eta(z) - \eta(X))]\,\delta(e(z) - e(X))\,dz$$

Since $y$ and $z$ are just integration variables, we can rename them and we bring over the second term on the LHS to the RHS. Thus we get, $\forall \eta, \forall X$ almost surely:

$$\int \left(\nabla_y \cdot [\|y - c(X)\|^2\,P_{\text{data}}(y)D_e^\top(y)(\eta(y) - \eta(X))]\right)\,\delta(e(y) - e(X))\,dy$$

$$= \frac{V(X)}{Z(X)}\int \left(\nabla_y \cdot [P_{data}(y)\,D_e^\top(y)(\eta(y) - \eta(X))]\right)\,\delta(e(y) - e(X))\,dy \tag{34}$$

In the above, we are dealing with divergences of the following form: $\nabla_y \cdot [f(y)M(y)v(y)]$, where $f_1(y) = \|y - c(X)\|^2 P_{\text{data}}(y)$ is a scalar (with $f_2(y) = \frac{V(X)}{Z(X)}P_{\text{data}}(y)$ on the other side), $M(y) =$

$D_e^\top(y)$ a $p \times k$ matrix, and $v(y) = \eta(y) - \eta(X)$ a $k$-vector. We will only expand the first term, namely:

$$\nabla_y \cdot [f(y)M(y)v(y)] = \nabla_y f(y) \cdot [M(y)v(y)] + f(y)\nabla_y \cdot [M(y)v(y)]$$

Eq. 34 thus decomposes into:

$$\int \nabla_y f_1(y) \cdot [M(y)v(y)]\, \delta(e(y) - e(X))\, dy + \int f_1(y)\nabla_y \cdot [M(y)v(y)]\, \delta(e(y) - e(X))\, dy =$$

$$\int \nabla_y f_2(y) \cdot [M(y)v(y)]\, \delta(e(y) - e(X))\, dy + \int f_2(y)\nabla_y \cdot [M(y)v(y)]\, \delta(e(y) - e(X))\, dy$$
(35)

We wish to go from an integral equality to a statement about the *integrands*. The argument will be made as follows:

1) We will argue that the second, divergence terms become negligible in the first-order stationarity conditions (Eq. 22).

2) *At this order* (i.e., at order $\varepsilon$), we will show that the *integrands* of the first terms must coincide *almost surely*.

From matching the second terms (with the perturbation inside the divergence), one might expect that on the level sets, we would have

$$f_1(y) \overset{a.e.}{\equiv} f_2(y),$$

which would lead to spherical level sets:

$$\|y - c(X)\|^2 = \frac{V(X)}{Z(X)},$$

which cannot be justified.

Another option is for the divergence $\nabla_y \cdot [M(y)v(y)]$ to vanish. The "trivial" solution $M(y)v(y) := D_e^\top(y)(\eta(y) - \eta(X)) = 0$ would imply that all perturbations are *tangential* to the level set; this would fly in the face of the variational argument where we allow $\eta$ to vary freely under a mild assumption (i.e., that it is differentiable and smooth).

Thus, we aim to show that the divergence $\nabla_y \cdot [M(y)v(y)]$ vanishes when integrated when considering the first-order optimality conditions. As a refresher, we have denoted the unperturbed level set (manifold) as:

$$L_{e(X)} = \{y \in \mathbb{R}^p : e(y) = e(X)\}$$

We have already assumed that $D_e$ has full row rank $k$ *a.e.* on $L_{e(X)}$. Thus, $\dim\left(L_{e(X)}\right) = p - k$. The normal space at $y \in L_{e(X)}$ is spanned by the rows of $D_e(y)$ (or, equivalently, the columns of $D_e^\top(y)$). Thus, any small displacement $\delta y \in \mathbb{R}^p$ of $y$ can be uniquely decomposed into the normal and tangential component:

$$\delta y = \delta y_\| + \delta y_\perp, \quad \text{with} \quad D_e(y)\, \delta y_\| = 0, \quad D_e(y)\, \delta y_\perp \neq 0$$

Let's consider now the perturbed level set for a small $\varepsilon$:

$$L_{(e+\varepsilon\eta)(X)} = \{y : (e + \varepsilon\eta)(y) = (e + \varepsilon\eta)(X)\}$$

As we've already shown: to the first order in $\varepsilon$, if $y$ is on $L_{(e+\varepsilon\eta)(X)}$, then we have Eq. 21:

$$e(y) - e(X) = -\varepsilon[\eta(y) - \eta(X)]$$

Set $y = y_0 + \delta y$ for some reference point $y_0 \in L_{e(X)}$, and expand around $y_0$:

$$e(y) - e(y_0) \approx D_e(y_0)(y - y_0)$$

Again, for $y$ on the (close-by) perturbed manifold, that difference must equal $-\varepsilon\left[\eta(y) - \eta(y_0)\right]$. So

$$D_e\left(y_0\right)\left[y - y_0\right] \approx -\varepsilon\left[\eta(y) - \eta\left(y_0\right)\right]$$

We have assumed that $\eta$ is Lipschitz, meaning that $\eta(y) - \eta\left(y_0\right)$ is $\mathcal{O}\left(\|y - y_0\|\right)$.

We now wish to show that the *normal component* of the displacement $\delta y = y - y_0$ is forced to be $\mathcal{O}(\varepsilon)$.

Denote the projection onto the row space of $D_e\left(y_0\right)$ as $\Pi_\perp$ (i.e., the projection into the normal space of the level set at $y_0$). We have:

$$D_e\left(y_0\right)\delta y = D_e\left(y_0\right)\left(\delta y_\| + \delta y_\perp\right) = D_e\left(y_0\right)\Pi_\perp \delta y$$

Thus:

$$D_e(y_0)\Pi_\perp \delta y = \Pi_\perp\left[-\varepsilon\left(\eta(y) - \eta\left(y_0\right)\right)\right] + \mathcal{O}(\varepsilon^2) = \mathcal{O}(\varepsilon)$$

Since $D_e(y_0)$ has full row rank by assumption, its pseudo-inverse $\exists$, hence we can state:

$$\Pi_\perp \delta y = \Pi_\perp\left[D_e(y_0)^\dagger\left(-\varepsilon(\eta(y) - \eta(y_0))\right)\right] + \mathcal{O}(\varepsilon^2)$$

Taking a norm on both sides:

$$\|\Pi_\perp \delta y\| = \|\Pi_\perp\left[D_e(y_0)^\dagger\left(-\varepsilon(\eta(y) - \eta(y_0))\right)\right]\| \le |\varepsilon|\,\|\Pi_\perp D_e(y_0)^\dagger\|\,\|(\eta(y) - \eta(y_0))\|$$
$$\le |\varepsilon|\,\|D_e(y_0)^\dagger\|\,\|(\eta(y) - \eta(y_0))\|,$$

where we repeatedly applied the Cauchy-Schwarz inequality and used the fact that the norm of a projection operator is one.

We will use the fact that both $e$ and $\eta$ are smooth and Lipschitz with constants $L_e$ and $L_\eta$. Since $D_e$ is $\mathcal{C}^1$ and has full row rank $k$ at $y_0$, its rank remains $k$ in a small open neighborhood of $y_0$ as rank can only change in discrete steps, and continuity prevents a drop for a small enough neigborhood. Consequently, the norm of the pseudo-inverse is also (locally) bounded by a finite constant $C$. Hence, we have

$$\|\delta y_\perp\| \le |\varepsilon|\,C\,L_\eta \|\delta y\| + \mathcal{O}(\varepsilon^2) = \mathcal{O}(\varepsilon),$$

i.e., any normal displacement from the original level set is $\mathcal{O}(\varepsilon)$.

Moving on to the divergence: we note that the divergence operator is a linear operator and can be decomposed into the tangential and normal component (relative to the current level set):

$$\nabla_y \cdot = \left(\nabla_{y,\|} \cdot\right) + \left(\nabla_{y,\perp} \cdot\right)$$

We have, $\forall \delta y$: $\nabla_{y,\|} \cdot \delta y_\perp = 0$ and $\nabla_{y,\perp} \cdot \delta y_\| = 0$

Since the terms we are taking the divergence over are of the form:

$$\nabla_y \cdot \left[D_e^\top(y)(\eta(y) - \eta(X))\right],$$

we thus have

$$\nabla_y \cdot \left[D_e^\top(y)(\eta(y) - \eta(X))\right] = \nabla_{y,\perp} \cdot \left[D_e^\top(y)\left(\eta(y) - \eta(X)\right)\right]_\perp$$

(since the column space of $D_e^\top$ spans exactly the normal space). Thus, only the normal components of the divergence will play a part.

Denote the vector field that the divergence is taken over as

$$u(y) = \left[M(y)v(y)\right]_\perp = \left[D_e^\top(y)[\eta(y) - \eta(X)]\right]_\perp$$

Namely, since $\|\delta y_\perp\| = \mathcal{O}(\varepsilon)$ (as shown above), we have again by Lipschitz-ness of the perturbation: $\|(\eta(y) - \eta(y_0))_\perp\| = \mathcal{O}(\|y - y_0\|) = \mathcal{O}(\|\delta y_\perp\|) = \mathcal{O}(\varepsilon)$. So $\|u(y)\| = \mathcal{O}(\varepsilon)$, since the norm of the Jacobian is bounded, as well, due to its regularity.

Next, let's define a "cylindrical" region $\mathcal{R}_\varepsilon$ around the old level set $L_{e(X)}$:

$$\mathcal{R}_\varepsilon := \left\{ y : \mathrm{d}\left(y, L_{e(X)}\right) \le c\,\varepsilon \right\}$$

for some small constant $c$ so that the perturbed level set is contained within $\mathcal{R}_\varepsilon$.

Finally, let's apply the divergence theorem:

$$\int_{\mathcal{R}_\varepsilon} \nabla_y \cdot u(y) dy = \int_{\partial \mathcal{R}_\varepsilon} u(y) \cdot \hat{n}(y) dS \le \|u\|_\infty \, \mathrm{Area}(\partial \mathcal{R}_\varepsilon)$$

$\|u(y)\|$ is $\mathcal{O}(\varepsilon)$ for all $y$ in the region, and the measure ("area") of $\partial \mathcal{R}_\varepsilon$ is at most $\mathcal{O}(\varepsilon)$ times the measure of the level set, as the "outer faces" of the "cylindrical surface" are $\mathcal{O}(\varepsilon^2)$. Hence the above integral is at most $\mathcal{O}(\varepsilon^2) \times$ the $p - k$-dimensional measure of $L_{e(X)}$. After accounting for the $f_1$ and $f_2$ factors in Eq. 35, we note:

1) $\left| \int_{\mathcal{R}_\varepsilon} f_{1,2}(y) \nabla_y \cdot u(y) dy \right| \le \sup_{y \in \mathcal{R}_\varepsilon} f_{1,2} \int_{\mathcal{R}_\varepsilon} |\nabla_y \cdot u(y)| \, dy$. As we're assuming $V(X)$ (Eq. 9) is finite, this means that $f_{1,2}$ is bounded. Hence the inclusion of the $f_{1,2}$ term doesn't change the $\mathcal{O}(\varepsilon^2)$ scaling of the flux.

2) If $L_{e(X)}$ were to extend to infinity (the "times the measure" part), the $P_{data}$ factor in $f_{1,2}$ would kill this contribution, as $P_{data}$ vanishes quickly enough for the level set variance $V(X)$ to be finite (by assumption).

Thus the (flux) integral is at most $\mathcal{O}\left(\varepsilon^2\right)$.

This means, that when considering the first-order optimality condition 22, for which we have obtained and expression of the form:

$$0 = \lim_{\varepsilon \to 0} \frac{1}{\varepsilon} (F(\varepsilon) - F(0)) = \left(\int \nabla_y f(y) \cdot \ldots dy \text{ - like terms}\right) + \left(\int f(y) \nabla_y \cdot \ldots dy \text{ - like terms}\right),$$

where we combine the terms from both sides of Eq. 35. We have shown that the second group of terms is $\mathcal{O}(\varepsilon^2)$, thus it vanishes in the limit and cannot play a role in the first-order optimality conditions.

Now to the second point. That is, *assume* that the second terms, where the variation $\eta$ appears inside the divergence, vanish in the first-order stationarity condition.

Thus we have the following integral equality, $\forall X, \forall \eta$:

$$\int \nabla_y \left[ \|y - c(X)\|^2 P_{\text{data}}(y) \right] \, D_e^\top(y)\, \delta(e(y) - e(X))\, (\eta(y) - \eta(X))\, dy$$

$$= \int \frac{V(X)}{Z(X)} \nabla_y \left[ P_{\text{data}}(y) \right] \, D_e^\top(y)\, \delta(e(y) - e(X))\, (\eta(y) - \eta(X))\, dy$$

Now, denote:

$$F_1(y, X) = \nabla_y \left[ \|y - c(X)\|^2 P_{\text{data}}(y) \right] \, D_e^\top(y)\, \delta(e(y) - e(X))$$

and

$$F_2(y, X) = \frac{V(X)}{Z(X)} \nabla_y \left[ P_{\text{data}}(y) \right] \, D_e^\top(y)\, \delta(e(y) - e(X))$$

So, the identity becomes:

$$\int F_1(y, X)\, (\eta(y) - \eta(X))\, dy = \int F_2(y, X)\, (\eta(y) - \eta(X))\, dy$$

While we have put the $\delta$ functions inside the integrands to be compared as to leave the (what are to be) test functions $\eta$ clearly separated, one needs to keep in mind that they will induce the integrals to be over the level set manifold surface measure. Additionally, $F_1$ and $F_2$ are now distributions. Also

note that we have assumed no specific constraints on $\eta$ besides them being smooth and Lipschitz — they are in the same function class as $e$ (cf. Assumption 2.3).

We will proceed via a proof by contradiction. Assume $\exists A \subseteq \{(y, X)\}$ with nonzero measure w.r.t. $dy\, dP_{data}(x)$ such that:

$$F_1(y, X)\mathbf{1}\{(y, X) \in A\} \neq F_2(y, X)\mathbf{1}\{(y, X) \in A\}$$

Furthermore, assume that $\pi_y(A) \subset \{y : e(y) = e(X)\}$, where $\pi_y$ is the projection to $y$. In other words, assume that the integrands differ on a "subsection" of the (unperturbed) level set (the delta functions inside $F$ would make the opposite — $A \not\subset L_{e(X)}$ — impossible, anyway).

We need to show that $\exists\, \eta$ s.t. the two integrals differ on a subset $S \in \mathbb{R}^p$ with $P_{data}(S) > 0$, as the integral identity is *a.s.* in $X$. To that end: $S = \pi_X(A) = \{X : \exists y \text{ s.t. } (y, X) \in A\}$ and $P_{data}(S) > 0$ as $A$ has non-zero mass under the joint measure.

Let's pick $\forall X \in S$ an $\eta_X$ to be a smooth function such that:

$$\eta(y) - \eta(X) = \begin{cases} 0, & \text{outside a small open neighborhood of } U_X \\ \text{nonzero and positive in all components} & \text{inside } U_X \end{cases},$$

where $U_X \Subset A \subset L_{e(X)}$ is an open set. That is, we're using the *bump function* and *partition of unity* approach common in calculus of variations [13; 9].

In this approach, we wish to build a global $\eta$ that belongs to the function class of the encoder. Pick a compact subset $S_K \subseteq S$ s.t. $P_{data}(S_K) > 0$. The compactness guarantees a finite sub-cover $\{U_{X_j}\}_{j=1}^m$. Then, we can construct a global perturbation $\eta$ by "gluing" together the bumps $\eta_X$ for that sub-cover using a smooth partition of unity $\rho_j(e(y))$. As we can select the cover to exclude the non-full-rank points of the encoder's Jacobian, this guarantees that the partition of unity $\rho_j(e(y))$ is smooth and exists. [3] Then, we define our "global" perturbation as:

$$\eta(y) = \sum_{j=1}^m \rho_j(e(y))\eta_{X_j}(y)$$

As $\eta$ is then a finite sum of smooth bumps multiplied by the smooth partition-of-unity functions, it is both $\mathcal{C}^1$ and Lipschitz.

Furthermore, we have for such $\eta$ *at least* one of the $\eta_{X_j}$ terms active in the integral $\forall X \in S_K$, leading to:

$$\int F_1(y, X)\,(\eta(y) - \eta(X))\, dy \neq \int F_2(y, X)\,(\eta(y) - \eta(X))\, dy$$

Since the integral equality must hold for any $\eta$, $\eta$-s form a rich function class, and our picked $\eta$ satisfies the requirements of the class, we arrive at a contradiction on a non-measure-zero set as $P_{data}(S_K) > 0$, thus it must hold $F_1(y, X) \overset{a.e.}{=} F_2(y, X)$.

Thus we have, in the first order of $\varepsilon$:

$$\nabla_y \left[ \|y - c(X)\|^2 P_{\text{data}}(y) \right]\, D_e^\top(y) \overset{\text{a.s. in } y}{=} \frac{V(X)}{Z(X)} \nabla_y \left[ P_{\text{data}}(y) \right]\, D_e^\top(y),$$

almost surely in $X$ on the level set of $e$.

Taking the gradients, we obtain:

$$\left[ 2(y - c(X))P_{\text{data}}(y) + \|y - c(X)\|^2\, \nabla_y P_{\text{data}}(y) \right]\, D_e^\top(y) \overset{\text{a.s. in } y}{=} \frac{V(X)}{Z(X)} \nabla_y P_{\text{data}}(y)\, D_e^\top(y),$$

and finally:

$$\frac{2(y - c(X))}{\frac{V(X)}{Z(X)} - \|y - c(X)\|^2}\, D_e^\top(y) \overset{\text{a.s. in } y}{=} \frac{\nabla_y P_{\text{data}}(y)}{P_{\text{data}}(y)}\, D_e^\top(y).$$

$\square$

---

[3] For the existence of the bump functions and the smooth partition of unity, see e.g. Lee [13], §2.

### A.2.3 Proof of Corollary 2.7

We adopt all the assumptions of Theorem 2.6. By the fact that we are at an extremum $y^*$, we have $\nabla_y P_{data}(y^*) = 0$. This gives us the following relation for Eq. 7:

$$\frac{2(y^* - c(X))}{\frac{V(X)}{Z(X)} - \|y^* - c(X)\|^2} \, D_{e^*}^\top(y^*) = 0.$$

$V(X)$ is finite (by assumption) and minimized on the level set due to the encoder's optimization objective 2, as is $Z(X)$, with additionally $Z(X) > 0$, since a smooth level set that contains an extremum must have some mass. Thus, the only way the denominator could go to $(-)$ infinity is if $\|y^* - c(X)\|$ was itself approaching infinity, which is not considered by assumption: this scenario represents "trivial" minima at distance approaching infinity.

As the denominator on the LHS cannot go to infinity, we must have:

$$(y^* - c(X)) \, D_{e^*}^\top(y^*) = 0.$$

This satisfied either by:

1) $(y^* - c(X)) = 0$, that is, the encoder aligns the level set so its center of mass coincides with the local extremum (most likely maximum), *or*

2) $(y^* - c(X)) \, D_{e^*}^\top(y^*) = 0$, that is, the normal projection of $(y^* - c(X))$ is exactly zero, meaning that $(y^* - c(X))$ lies in the tangent space of the level set at the extremal point.

$\square$

## A.3 Proofs of Section 3

### A.3.1 Comment: DPA recovers the data distribution

To refresh: for an optimal encoder/decoder pair, we have by definition:

$$d^*\left(e^*(X), \epsilon\right) \sim P_{e^*, X}^*$$

That is:

$$d^*(z, \epsilon) \overset{d}{=} (X | e^*(X) = z), \quad \forall z$$

Treating $Z = e^*(X)$ as a random variable, we have:

$$d^*(Z = z, \epsilon) \overset{d}{=} (X | Z = z)$$

Thus, for any measurable set $A$:

$$\mathbb{P}(X \in A) = \int P(X \in A \mid Z = z) \, P_Z(z) \, dz$$

and

$$\mathbb{P}(d^*(Z, \epsilon) \in A) = \int P(d^*(Z, \epsilon) \in A \mid Z = z) \, P_Z(z) \, dz$$

Since $d^*(Z = z, \epsilon) \overset{d}{=} (X | Z = z)$, the two integrals match, and by the law of total probability:

$$d^*(e(X), \epsilon) \overset{d}{=} X \tag{36}$$

### A.3.2 Proof of Propositions 3.5 and 3.2

We assume that the manifold $\mathcal{M}$ can be exactly parameterized by a smooth, injective $e$ in the first $K$ components $e_{1:K} : \mathcal{M} \to \mathbb{R}^K$, and that our encoder function class is expressive enough to achieve this.

If we assume this $e_{1:K}$ is optimal among $K$-dimensional encoders, this must necessarily imply by the definition of the ORD:

$$d^* \left( [e_{1:K}(x), \epsilon_{(K+1):p}] \right) \sim P^*_{e_{1:K,x}}$$

where $d^*$ is the accompanying optimal decoder.

Now, due to the injectivity of the parameterization, we must have for the RHS: [4]

$$P^*_{e_{1:K,x}}(x) = \text{Law}(X|e(X) = e(x)) = \delta(x), \; \forall x. \tag{37}$$

In other words, we get an almost-sure equality:

$$d^* \left( e_{1:K}(X), \epsilon \right) \overset{\text{a.s.}}{=} X$$

This implies for the two terms in $L_K[e_{1:K}, d^*]$ (Eq. 11):

$$\mathbb{E}_X \mathbb{E}_{Y \sim P_{d, e_{1:K}(X)}} \left[ \|X - Y\|^\beta \right] = \mathbb{E}_X \mathbb{E}_{Y \sim \delta(X)} \left[ \|X - Y\|^\beta \right] \equiv 0$$

and

$$\mathbb{E}_X \mathbb{E}_{Y, Y' \overset{\text{iid}}{\sim} P_{d, e_{1:K}(X)}} \left[ \|Y - Y'\|^\beta \right] = \mathbb{E}_X \mathbb{E}_{Y, Y' \overset{\text{iid}}{\sim} \delta(X)} \left[ \|Y - Y'\|^\beta \right] \equiv 0$$

Since $L_k[e, d] \geq 0$, this is indeed the global minimum. Thus, the encoder is an $K$-best-approximating encoder, and the manifold is $K'$-parameterizable by $(e_{1:K}, d^*)$ with $K' = K$.

$\square$

### A.3.3  Proof of Theorem 3.4

We consider the setting where the data $X$ are still supported on a $K$-dimensional manifold, which, however, might not be globally parameterizable by a single, smooth encoder $e$ with $K$ output dimensions.

We have defined the $K'$-best-approximating encoder as the encoder that minimizes the loss the when considering terms up to the $K'$-th term only:

$$(e^*, d^*) \in \underset{e,d}{\arg\min} \sum_{k=0}^{K'} L_k[e, d]$$

again taking the weights to be uniform, *and* which achieves the globally best energy score in its $K'$-th term:

$$L_{K'}[(e^*, d^*)] = \min_{e,d,k} L_k[e, d].$$

This also automatically makes the manifold $K'$-parameterizable, with any remaining manifold variance being non-explainable with a DPA.

By the fact that for $\beta \in (0, 2)$, the energy score is strictly proper, this encoder/decoder pair is indeed *the unique* global optimum for $K'$-dimensional encoders, with $P_{d^*, e^*_{1:K'}(X)}$ being the *unique* distribution minimizing $L_{K'}[(e^*, d^*)]$.

Now consider the overall $p$-dimensional problem given by Eq. 10. The terms for $K' + 1 \dots p$ cannot do better than $L_{K'}[e^*, d^*]$ by the definition of our $K'$-best-approximating encoder, thus the best an encoder can do is to output the same distribution $P_{d^*, e^*_{1:K'}(X)}$.

Next we observe that the $K'$-dimensional optimization is a nested subproblem of the $p$-dimensional one. Thus, when optimizing over $p$ dimensions, the optimal encoder must coincide with the above $K'$-best-approximating encoder in terms up to $K'$, otherwise it would contradict the latter being the global optimum among $K'$-dimensional encoders: one would obtain a lower loss by simply using the first $K'$ components of the $p$-dimensional encoder.

Hence, the $K'$-best-approximating encoder is the global optimum for all $p$ dimensions and we obtain for the optimal solution:

$$P_{d^*, e^*_{1:k}(X)} = P_{d^*, e^*_{1:K}(X)}, \; \text{ for } k > K'$$

---

[4] This has been given as an example on page 6 of Shen and Meinshausen [18] for a general invertible $e^*$.

Next, we wish to show that the "extra" dimensions are independent of the data conditioned on the relevant components $1 : K'$. Consider the $(K' + 1)$-th component. Note that $e^*_{1:K'}(X)$ and $e^*_{1:K'+1}(X)$, when viewed as joint distributions (over dimensions of $e^*$), form a filtration; denote $\mathcal{F}_K = \sigma((e^*_1(X), \ldots, e^*_{K'}(X))) \subseteq \mathcal{F}_{K'+1} = \sigma((e^*_1(X), \ldots, e^*_{K'+1}(X)))$.

By Eq. 13, we have

$$X|\mathcal{F}_{K'+1} \stackrel{d}{=} X|\mathcal{F}_{K'},$$

hence

$$\mathbb{E}[f(X)|\mathcal{F}_{K'+1}] = \mathbb{E}[f(X)|\mathcal{F}_{K'}]$$

for any (Borel-measurable) function $f$.

Let $Z$ be $\mathcal{F}_{K'+1}$ — but not $\mathcal{F}_{K'}$ — measurable. The claim from the theorem is then equivalent to the following conditional independence:

$$Z \perp\!\!\!\perp X|\mathcal{F}_{K'}$$

or

$$\mathbb{E}[g(Z)f(X)|\mathcal{F}_{K'}] = \mathbb{E}[g(Z)|\mathcal{F}_{K'}]\,\mathbb{E}[f(X)|\mathcal{F}_{K'}]$$

for any pair of integrable functions $f, g$.

We can prove this claim in the following way:

$$\mathbb{E}[g(Z)f(X)|\mathcal{F}_{K'}] \underbrace{=}_{\text{tower}} \mathbb{E}[\mathbb{E}[g(Z)f(X)|\mathcal{F}_{K'+1}]|\mathcal{F}_{K'}] \underbrace{=}_{Z \text{ is } \mathcal{F}_{K'+1} \text{ meas.}} \mathbb{E}[g(Z)\,\mathbb{E}[f(X)|\mathcal{F}_{K'+1}]|\mathcal{F}_{K'}]$$

$$\underbrace{=}_{\mathbb{E}[f(X)|\mathcal{F}_{K'+1}]=\mathbb{E}[f(X)|\mathcal{F}_{K'}]} \mathbb{E}[g(Z)\,\mathbb{E}[f(X)|\mathcal{F}_{K'}]|\mathcal{F}_{K'}] \underbrace{=}_{\mathbb{E}[f(X)|\mathcal{F}_{K'}]\,\mathcal{F}_{K'}\text{ meas.}} \mathbb{E}[g(Z)|\mathcal{F}_{K'}]\,\mathbb{E}[f(X)|\mathcal{F}_{K'}]$$

The other dimensions $K' + 2, \ldots, p$ follow by the fact that the filtrations are nested, meaning that the above derivation holds when $\mathcal{F}_{K'+1}$ is replaced by $\mathcal{F}_{K'+2}$, etc. Thus, we have shown that the "extra" dimensions are independent of the data, given the relevant first $K'$ components.

$\square$

### A.3.4    Discussion on Remark 3.6

We are considering the "exactly-parameterizable" manifold scenario. We have shown in Proof A.3.2 that the $K$-th terms $L_K[e, d]$ are identically zero and thus the global minimum.

As in Proof A.3.3, we observe for the higher terms (e.g., $K + 1$-th):

$$\mathbb{E}_X \mathbb{E}_{Y \sim P_{d^*, e^*_{1:K+1}(X)}} \left[ \|X - Y\|^\beta \right] > 0$$

and

$$\mathbb{E}_X \mathbb{E}_{Y,Y' \stackrel{\text{iid}}{\sim} P_{d^*, e^*_{1:K+1}(X)}} \left[ \|Y' - Y\|^\beta \right] > 0$$

*unless*

$$P_{d^*, e^*_{1:K+1}}(X) = \delta(X) = P_{d^*, e^*_{1:K}}(X)$$

In other words, the only zero variance distribution is the delta distribution, which is (by assumption) the distribution induced by $P_{d^*, e^*_{1:K}}$. Thus, the encoder that parameterizes the manifold in the $e_{1:K}$ dimensions and outputs the same distribution for terms $K + 1 : p$ is *the* optimal encoder for the terms $K : p$. We will keep $e^*$ to denote this encoder in the following argument.

Next, we argue that *typically* for $p \gg K$, this is also the optimal encoder for the first $K - 1$ dimensions, thus the $K$-best-approximating one, or equivalently, the global optimum.

Again, assume that all the weights $\omega_k \in [0, 1]$ are uniform, i.e. $\frac{1}{p+1}$. As discussed in Shen and Meinshausen [18], it remains an open question whether an optimal encoder is necessarily the one that minimizes all the terms in the loss *simultaneously* (which is the case for the terms $K : p$ when the encoder is the $K$-best-approximating one), so the following argument will examine what is *likely* to happen for parameterizable manifolds as $p \gg K$.

Suppose that there is another, different (in the sense that it reconstructs different ORDs) globally optimal $(\tilde{e}, \tilde{d})$ pair that, by "sacrificing" perfect reconstruction at dimensions $K : p$, improves on

$L_k(e^*, d^*)$ for terms $k = 1, \ldots, K-1$ to such an extent as to strictly beat the manifold-parameterizing encoder $e^*$ in the aggregate loss. Note that for the latter, the "partial" manifold parameterization should *typically* (i.e., for non-pathological manifolds) already be a reasonably good encoding for dimensions $k < K$ (in the energy score sense), so $L_k[e^*, d^*]$ is *unlikely* to be very far from the minimum for each of these $k$.

We will be using Proposition 1 (together with Theorem 1) of [18], which states that at the optimum, the two terms in the loss are equal and thus focusing on the reconstruction one; as we are conjecturing the existence of another global *optimum*, this is valid.

Since the data manifold is $K$ dimensional, the encodings for $k < K$ cannot describe the manifold, leading to imperfect reconstruction: $\|X - Y\|^\beta > 0$. This is true for any encoder/decoder pair, and we can denote the *global* minimal loss when considering $k$ dimensions (i.e.. a single term in the optimization objective) as:

$$R_{k,\min} := \min_{e_{1:k}, d} L_k[e, d] > 0$$

Furthermore, *typically* for $k < K$ each encoding has to "describe" $K - k$ additional "directions" of the manifold, meaning that $P^*_{e_{1:k}, X} = (X| \; e_{1:k}(X) = z)$ becomes more spread out on the manifold as $k$ decreases; hence, *typically*: [5]

$$R_{k,\min} \geq R_{k',\min} \text{ for } k < k',$$

which implies

$$L_k[\tilde{e}, \tilde{d}] \geq L_{k'}[\tilde{e}, \tilde{d}] \text{ for } k < k'$$

for any reasonable (assumed-to-be) optimal $\tilde{e}$ as the manifold's dimension is $K > k' > k$.

Suppose now that for some $S \subseteq \{1, \ldots, K - 1\}$ this pair reduces the loss $L_k[\tilde{e}, \tilde{d}]$ below $L_k[e^*, d^*]$, which implies (by "breaking" the manifold parameterization):

$$L_k[\tilde{e}, \tilde{d}] > L_k[e^*, d^*] = 0, \; \forall k \geq K$$

Denote the hypothesized improvements (in absolute value) on the $k < K$ terms as $\Delta_k$, and the latter costs for $k \geq K$ as $\epsilon_k$:

$$\Delta_k := L_k[e^*, d^*] - L_k[\tilde{e}, \tilde{d}] \quad k < K, \quad (> 0 \text{ if there is an improvement})$$

$$\varepsilon_k := L_k[\tilde{e}, \tilde{d}] - L_k[e^*, d^*] \quad k \geq K \quad (> 0).$$

Thus we have the following best-case trade-off that needs to hold in order to $(\tilde{e}, \tilde{d})$ to be optimal:

$$(K - 1) \cdot \max_k \Delta_k > (p - K) \min_k \epsilon_k \tag{38}$$

$\epsilon_k$ are nonzero as $\delta$ is the only zero-variance distribution:

$$\epsilon_k = L_k[\tilde{e}, \tilde{d}] - L_k[e^*, d^*] \geq c > 0,$$

and the improvements $\Delta_k$ are bounded above by:

$$\Delta_k \leq L_k[e^*, d^*] - \underbrace{R_{k,\min}}_{>0}, \quad k < K.$$

As discussed, *typically*, $L_k[e^*, d^*]$ are unlikely to be catastrophically large as the partial manifold parameterization should do reasonably well on the energy score loss. Denoting:

$$M := \max_{k<K} L_k[e^*, d^*], \quad m := \min_{k<K} R_{k,\min},$$

we can express Eq. 38 as:

$$(K - 1)(M - m) > (p - K) \, c$$

As the LHS is a constant for non-pathological manifolds and the RHS is a linear function of $p$, Eq. 38 cannot hold as $p$ increases. Stated more formally:

---

[5]While there might be other pathological scenarios, in most cases this would imply the true manifold dimension to be $K < k'$, which contradicts the assumption.

**Lemma A.1 (The parameterizing encoder is typically globally optimal)**
*Fix $M, m, c > 0$ as noted above. The manifold-parameterizing encoder $e^*$ is the unique global minimizer whenever:*

$$p > \left( \frac{K-1}{c} \right) (M - m) + K. \tag{39}$$

Thus, *typically*, an encoder that parameterizes the manifold in the first $K$ dimensions is an optimal encoder. Then, due to Prop. 3.5, the results in Theorem 3.4 follow.

# B Experimental Appendix

## B.1 Data distributions for the experiments

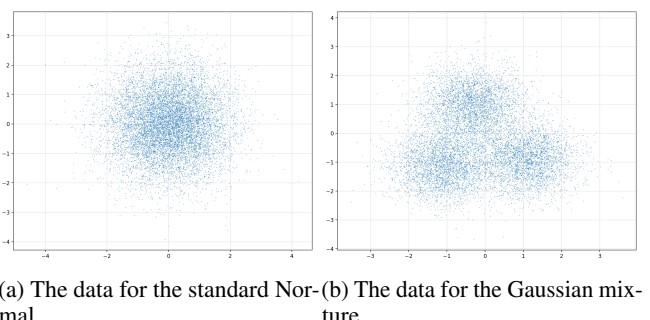

(a) The data for the standard Nor-(b) The data for the Gaussian mix-
mal.                                 ture.

Figure 3: The data for the Gaussian examples.

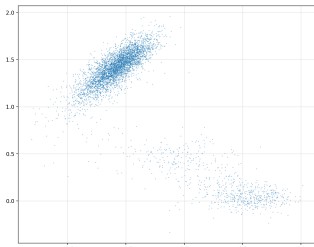

Figure 4: The data for the Müller-Brown potential example. Note the dearth of samples in between the potential minima.

## B.2 Score alignment

Our method of extracting level-sets is relatively rough and uses `matplotlib`'s `contour` function. Thus, there are significant inaccuracies in estimating the level set statistics (e.g., $c(X)$), prompting us to discard scalar factors on both sides of Eq. 7, and evaluate *alignment* instead. Such inaccuracies are especially impactful around the *critical shell*, where a slight numeric error can erroneously flip the direction of the vector on the LHS of Eq. 7, as illustrated in Fig. 5. As is evident in the figure, the *directional alignment* remains very good. To account for these drawbacks, we opted for *absolute* cosine similarity as a measure of the match between the level-set geometry and the score as predicted by Theorem 2.6.

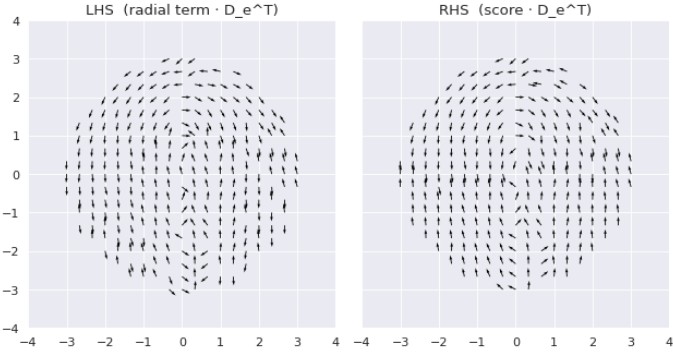

Figure 5: Signed score alignment: sign flips due to the inaccuracy of estimating the level-set statistics.

### B.3  Minimum Free Energy Path further results and discussion

In the use of unsupervised learning methods in computational chemistry, one typically wishes to find a good *collective variable* [4]: a representation that can approximate the important transitions between physical states. For the Müller–Brown potential, the states are represented as minima of the potential energy. The ideal collective variable is, naturally, the minimum free-energy path (MFEP), as traveling along the latter is the least-costly way for the system to undergo transitions. Thus, the better a method is able to parameterize the latter in a single component, the better this encoding can serve in downstream tasks, such as *enhanced sampling* [4], where the encoding is used to modify the potential to "guide" the system towards undergoing transitions between states.

As discussed in the main sections, the fact that the DPA aligns the level sets with the force field naturally induces an approximate parameterization of the MFEP, as changing the encoding value represents moving to the closest level set, and this move is aligned with the score – the force field. In this section, we present further results of the automated experiment used to generate Table 2.

In the experiment, we first train the encoders on the data (cf. Fig. 4) across 25 different random seeds. Next, we encode the MFEP, obtained by the string method [8]. We find the best encoding component by regressing the latter against the cumulative arc length of the MFEP and picking the component with the largest $R^2$ coefficient. If the encoding is the exact parameterizing parameter of the MFEP, the latter should be almost 1. Next, we select the range of the encodings to travel over to obtain our path approximation by finding the closest encoding points to the start and the end of the MFEP by using the same regression. With this, we travel over the range of the encodings and, at each step, predict the next position $x$ by finding the point corresponding to the next encoding in the range via a root-finding algorithm; that is, by inverting the encoder: $x_{next} = e^{-1}(z_{next})$. Additionally, we observe that for the DPA, the best encoding is *always a monotonically increasing* (approximate) parameterization of the MFEP, and the level sets are connected. Hence, we also present an alternate method, where we generate the MFEP approximation by simply taking a step in the direction of the gradient of the encoder, obtained by automatic differentiation. The results essentially coincide with the more involved root-finding results, and are used in Fig. 6 together with the results obtained from root-finding for the other model architectures.

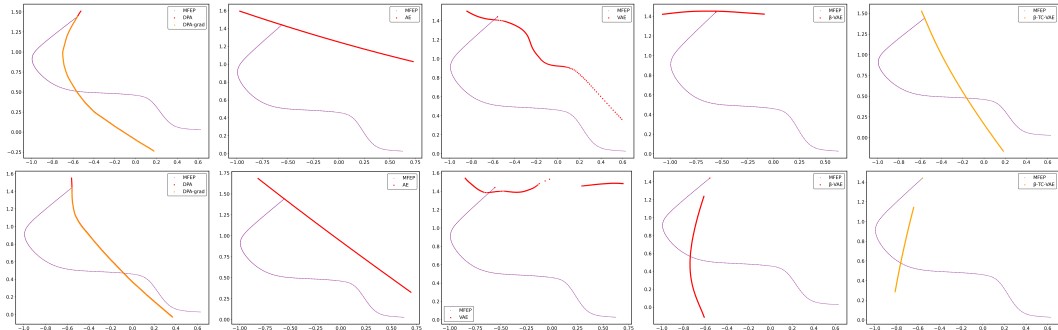

Figure 6: Best MFEP parameterizations for *Left-to-right:* DPA, Autoencoder, VAE, $\beta$-VAE, $\beta$-TCVAE for two arbitrarily selected random seeds: *top*: 43, *bottom:* 63.

The results consistently demonstrate that the DPA performs much better as a scalar parameterizer of the MFEP; much tighter approximations can be obtained (without retraining the model) by decreasing the step size (in the latent $z$) and manually adjusting the range of the encoding, as illustrated in Fig. 7. That is, the method of obtaining the approximate paths is relatively crude, but unbiased with regard to the model architectures. As noted in the main work, the Autoencoder typically obtains the correct direction between the first and last minimum, while the VAE does not and performs worse, with the level sets often being disconnected, as can be observed in the bottom row of the figure.

Below, we summarize the metrics used in Table 2, where we present the results obtained from 25 runs with different random seeds. We retrain the models on each run and find the best parameterization of the MFEP possible by the encoder.

- *Average closest distance from the parameterizing path to the MFEP* $\bar{d}_{\text{path} \to \text{MFEP}}$.
- *Average closest distance from the MFEP to the parameterizing path* $\bar{d}_{\text{MFEP} \to \text{path}}$.

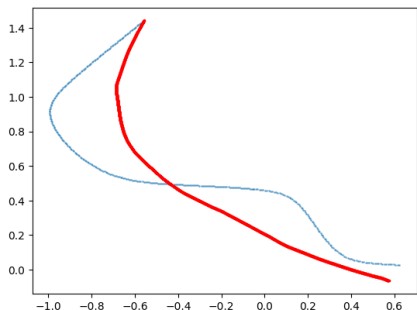

Figure 7: Tighter approximation of the MFEP for the DPA.

- *The Chamfer distance $\bar{d}_{\mathrm{C}}$*: the average of the above two distances.
- *The Hausdorff distance $\bar{d}_{\mathrm{H}}$*: the maximum distance between either paths.
- *$95^{th}$-percentile error*: similar to the Hausdorff distance, we present the larger of the two $95^{th}$-percentiles for the two distance sets.

Additionally, we examine which encoder component was the closest parameterization — denoted as $z_\parallel$ — and present the average over the random seeds. Of note here is the principal-components aspect of the DPA, as the *first* component always represents the closest parameterization, as the MFEP is the single most important feature for describing the Müller–Brown potential. This does not hold for other models.

### B.4 Independence of extraneous latents

Table 5: Datasets used for the extraneous–latent independence experiments (Sec. 4.2). $p$ is ambient dimension; $K$ is intrinsic dimension. "Generating map / source" summarizes the construction.

| Dataset | p | K | Generating map / source (brief) |
| --- | --- | --- | --- |
| Gaussian line | 2 | 1 | Line embedded in $\mathbb{R}^2$ with orthogonal Gaussian noise (1-D signal + small normal perturbation). |
| Parabola | 2 | 1 | Curve $(t,\ t^2)$ sampled over a bounded interval. |
| Exponential | 2 | 1 | Curve $(t,\ \exp(t))$ sampled over a bounded interval. |
| Helix slice | 3 | 2 | Helical sheet $(\cos t,\ \sin t,\ t)$ (thin slice of a 3D helix). |
| Grid sum | 3 | 2 | $(x, y, z)$ with constraint $z = x + y$ (two free coordinates; one dependent). |
| S-curve | 3 | 2 | Standard `sklearn` S-curve surface in $\mathbb{R}^3$ (two-dimensional manifold). |
| S-curve ($\beta = 2$) | 3 | 2 | Same S-curve as above; DPA trained with $\beta = 2$ to show empirical robustness. |
| Swiss-roll (3D) | 3 | 2 | Standard `sklearn` Swiss-roll surface in $\mathbb{R}^3$ (two-dimensional manifold). |

**Details about determinism checks:** For the regression part, we fit the following nonparametric regressors: cubic B-splines with up to 64 knots, random forests with 400–1000 trees, or high-degree polynomials. For the conditional entropy $H(U \mid Z) = H(U, Z) - H(Z)$ we use $k$-NN Kozachenko–Leonenko estimation with $k = 5$.

**Details about CIT:** We adopt a *double Conditional Randomization Test* (CRT). We (i) fit a Gaussian process to model the conditional distribution of $U$ given $Z$ obtaining the mean $\mu(Z)$ and variance $\sigma^2(Z)$; (ii) compute the observed test statistic as the HSIC (Hilbert-Schmidt Independence Criterion) between $U$ and $X$; (iii) generate $B$ bootstrap null samples by drawing $U_b \sim gN(\mu(Z), \sigma^2(Z))$ and decoding the latent representation $[Z, U_b]$ through a trained decoder to obtain synthetic data $X_b$; (iv) compute HSIC statistics for each bootstrap sample; and (v) calculate the p-value as the proportion of bootstrap statistics that exceed the observed statistic. The test is repeated $N$ times with fresh decoder samples to assess the validity of the test under the null hypothesis that $U \perp\!\!\!\perp X \mid Z$.

**Additional diagnostics on the S-CURVE**

We train DPA with $k = 3$ on the S-CURVE dataset ($K = 2$ intrinsic dimension). We report unconditional and conditional (given $Z_{1:2}$) distance correlation and mutual information between each latent $Z_i$ and the data $X$, plus a conditional linear $R^2$ summary. For distance correlation we use the `dcor` package; its *partial* estimator is used as a proxy for conditional independence (near-zero values are consistent with $Z_i \perp\!\!\!\perp X \mid Z_{1:2}$. For mutual information we use `sklearn.feature_selection.mutual_info_regression`; we report the "per-coordinate maximum" $\text{MI}_{\max} = \max_j I(Z_i; X_j)$, and its conditional analogue after residualizing both $Z_i$ and $X_j$ on $Z_{1:2}$ with kernel-ridge regression. This yields a conservative check for residual dependence. All metrics are computed on a held-out test set after standardizing $X$ and $Z$.

Table 6: S-curve diagnostics ($N = 10{,}000$, $K = 2$). "cond." conditions on $Z_{1:2}$.

| Latent | Distance corr. | | Mutual info. (nats) | | $R^2$ (linear, cond.) |
|---|---|---|---|---|---|
| | uncond. | cond. | uncond. | cond. | value |
| $Z_1$ (relevant) | 0.71 | — | 1.49 | — | — |
| $Z_2$ (relevant) | 0.57 | — | 0.67 | — | — |
| $Z_3$ (extra) | 0.27 | **0.07** | 0.29 | **0.13** | **0.0006** |

As a sanity check on the intrinsic dimension, conditioning on only one latent ($K = 1$ hypothesis) leaves substantial linear dependence: $R^2(X; Z_2 \mid Z_1) = 0.94$; under the correct $K = 2$ hypothesis, $R^2(X; Z_3 \mid Z_{1:2}) = 0.0006$. Together with the main tests in Sec. 4.2, these complementary diagnostics support the uninformativeness of the extraneous latents result.

## B.5 Experimental Details

### B.5.1 Experiment code

The code required to reproduce the results is provided at github.com/andleb/DistributionalAutoencodersScore. Please refer to README.md document therein for full details on the code structure and details.

The hyperparameter choices, the optimizer types, etc. used for generating the results are provided in Sec. B.5.2 below.

### B.5.2 Experimental details

**Figure 1 (Gaussian Score):** Uses 2D standard Gaussian data with analytical score function. Random seed: 42. DPA models trained with $\beta = 2$, deterministic encoder, stochastic decoder, 3 latent dimensions ($k = 3$), 4-layer networks with 256 hidden units per layer, residual blocks enabled. Training used standardized inputs.

**Table 1 (Score Alignment):** Evaluates cosine alignment between $\nabla_{\mathbf{x}} \varphi_k(\mathbf{x})$ and $\nabla_{\mathbf{x}} \log p(\mathbf{x})$ on a $100 \times 100$ grid over $[-4, 4]^2$. Tests two distributions: (i) 2D standard Gaussian and (ii) trimodal Gaussian mixture with means $\mu_1 = (-1.1, -1.1)$, $\mu_2 = (1.1, -0.9)$, $\mu_3 = (-0.33, 1.0)$, equal weights $w_j = 1/3$, and shared variance $\sigma^2 = 0.66^2$. Density cutoff: 1% of maximum density. Reports mean $|\cos \theta|$, standard deviation, and 95th percentile across grid points with sufficient density.

**Figure 2 (Müller-Brown Potential):** Uses Müller-Brown potential energy surface with standard parameters. Collective variables learned via DPA with $\beta = 2$, 2D latent space ($k = 2$), 4-layer networks with 100 hidden units. Training trajectories generated via molecular dynamics at temperature $T = 300$K. Random seed: 42.

**Table 2 & Figures 6–7 (MFEP Comparisons):** Compares five autoencoder variants on minimum free-energy path (MFEP) parameterization for the Müller-Brown potential: (i) DPA ($\beta = 2$, deterministic encoder, stochastic decoder), (ii) standard autoencoder (AE), (iii) vanilla VAE, (iv) $\beta$-VAE ($\beta = 4$, Higgins loss), and (v) $\beta$-TC-VAE ($\alpha = 1$, $\beta = 6$, $\gamma = 1$). Each method trained across

$N = 25$ random seeds (42–66). Training data: 10,000 Müller-Brown MD samples at $T = 300$K. All models use 2D latent space, encoder/decoder with two hidden layers of 100 units, ELU activations. Training: 1200 epochs, Adam optimizer with lr $= 10^{-3}$ (VAE variants) or $5 \times 10^{-4}$ (DPA), batch size 5000 (DPA/AE/VAE) or 256 (TC-VAE to ensure batch $\geq 2$ for log-density-ratio estimation). Reference MFEP computed via string method with 32 nodes, tolerance $10^{-7}$. For each model, summary statistics computed by dropping the worst-performing seed (by Chamfer distance) and reporting mean $\pm$ standard deviation over remaining 24 seeds.

**Table 3 (Conditional Independence – Deterministic Tests):** DPA models: $\beta \in \{1, 1.5, 2\}$, $k = 1$ informative + 1 extraneous latent, 4-layer encoder/decoder with 100 hidden units, batch size 128, trained for 1000 epochs. Diagnostics: (i) $R^2$ via cubic B-splines (up to 64 knots), random forests (400–1000 trees), or high-degree polynomials; (ii) intrinsic dimension via Levina–Bickel MLE with 200 bootstrap samples; (iii) conditional entropy $H(U \mid Z)$ via $k$-NN Kozachenko–Leonenko estimator ($k = 5$). Random seed: 42.

**Section 4.2 (Conditional Randomization Test):** Double CRT with $B = 500$ bootstrap samples, $N_{\text{reps}} = 50$ repetitions. Gaussian process fits conditional $U_2 \mid Z_1$ using RBF kernel. Test statistic: HSIC with Gaussian kernels (median heuristic for bandwidth). Null samples drawn as $U_b \sim \mathcal{N}(\mu(Z_1), \sigma^2(Z_1))$ and decoded via trained decoder. Significance level $\alpha = 0.05$.

### B.5.3 Existing assets

The experimental code makes use of modified routines from the **mlcolvar** project [4]: `https://github.com/luigibonati/mlcolvar`, available under the MIT license. Additionally, the data for the Müller–Brown potential examples is bundled with the examples provided in the `mlcolvar` repository, and reproduced in this supplement for convenience.

The original license texts are kept verbatim in `third_party_licenses/<project>/LICENSE`. All other dependencies are installed via `pip` (cf. the `requirements.txt` file provided); their names,versions and licences appear in `third_party_licenses/THIRD_PARTY_LICENSES.md`.

### B.5.4 Compute resources

The results presented in the main paper were run on a laptop and required about 10 minutes for each model training and about 20 seconds for the detailed plots.

The MFEP parameterization experiment was run on a single Nvidia V100 GPU, taking 1 hour and 38 seconds of wall time. The Independence experiments took roughly 5 hours on a single Nvidia V100 GPU.

### B.6 LLM usage

We used GPT-4 interactively as a search tool and sanity checker for content created by the authors. All LLM-generated suggestions were independently verified against the original literature.

