# OpenReview forum: "Distributional Autoencoders Know the Score"
_NeurIPS.cc/2025/Conference — NeurIPS 2025 poster_

### Official Review · Reviewer_YCG7 · 2025-06-27

**Clarity:** 3
**Significance:** 2
**Originality:** 3
**Rating:** 4
**Confidence:** 3

**Summary:**

This paper provides a rigorous theoretical analysis of the Distributional Principal Autoencoder (DPA), a model that combines principled distributional reconstruction with interpretable latent representations. The authors derive an exact closed-form relation connecting the encoder’s optimal level sets to the data distribution’s score function, offering a geometric understanding of how the model captures the data structure. They further prove that for data lying on a low-dimensional manifold, the latent dimensions beyond the manifold’s intrinsic dimension become conditionally independent and carry no additional information, enabling precise intrinsic dimension estimation without extra regularization. The theory is supported by experiments on synthetic distributions and the Müller-Brown potential.

**Questions:**

$\textbf{Questions for the Authors:}$

1) The theoretical analysis assumes that the encoder and decoder achieve global optimality in minimizing the DPA loss (e.g. Definition 3.1., and the related theoretical results Proposition 3.2, Theorem 3.4). Given the non-convexity of neural network training and the common presence of local minima, how realistic is this assumption in practice?

2) The theory requires the data manifold to be exactly or approximately parameterizable by the encoder’s function class. How strong is this assumption, and how does it hold up when modeling complex or noisy real-world data?

3) The experiments primarily involve low-dimensional synthetic data and the Müller-Brown potential. How well do these empirical findings generalize to complex, high-dimensional datasets, such as MNIST, climate simulations, or single-cell genomics, which were explored in the original DPA paper?

4) The main results rely on population-level optimality, assuming infinite data and exact minimization of the DPA loss. Could the authors clarify theoretically or empirically how finite samples and imperfect training affect the level set alignment (Theorem 2.6) and the disentanglement of extraneous dimensions (Theorem 3.4)?

5) The results assume that the encoder’s Jacobian has full row rank almost everywhere on the level sets. How restrictive is this assumption in practice?

**Ethical Concerns:**

["NO or VERY MINOR ethics concerns only"]

**Final Justification:**

The discussion on global optimality and manifold parameterizability helped me clarify my earlier misunderstanding regarding the assumptions behind Theorem 3.4 and Corollary 3.5.

**Limitations:**

Yes.

**Paper Formatting Concerns:**

No concerns.

**Quality:**

3

**Strengths And Weaknesses:**

$\textbf{Strengths}$

$\textbf{Quality:}$
The paper presents mathematically precise, non-asymptotic results connecting the DPA encoder’s level sets to the data score function. The derivations provide a solid framework explaining the DPA’s latent space geometry. Additionally, by proving that extraneous latent dimensions are conditionally independent, the work offers a novel formalization of intrinsic dimension estimation without extra regularization or architecture changes.

$\textbf{Clarity:}$
The paper clearly defines key concepts. The logical flow from assumptions to lemmas, theorems, and corollaries is coherent and supports the main narrative effectively.

$\textbf{Significance:}$
The connection between encoder geometry and the data score may inspire new generative model architectures and more interpretable latent variable models.

$\textbf{Originality:}$
While building on the existing DPA model, the exact characterization of level sets in terms of score functions and conditional independence results for extraneous dimensions appear original.


$\textbf{Weaknesses}$

The theoretical guarantees provided by the DPA rest on several strong assumptions that may limit their applicability in practice. First, the results require that the encoder and decoder achieve global optimality in minimizing the DPA loss, an assumption that is difficult to satisfy due to the non-convex nature of deep network training and the potential for convergence to suboptimal local minima. Second, the encoder function class must be sufficiently expressive to realize an exact parameterization of the underlying data manifold, which may not hold for networks with limited capacity or when faced with highly complex or irregular data structures. Finally, the data distribution is assumed to lie exactly or approximately on a low-dimensional manifold, an idealization that real-world datasets often violate due to noise, outliers, or intrinsic complexity.

Consequently, while the theoretical framework offers valuable insight into the capabilities of DPA under ideal conditions, translating these guarantees into practical success requires careful consideration of these assumptions and further empirical validation. The empirical validation is also limited to low-dimensional synthetic examples and a classical benchmark (Müller-Brown potential), which may not fully demonstrate the model’s robustness or scalability to more complex, high-dimensional real-world data. For example, the original DPA paper evaluated the model on diverse datasets including MNIST, synthetic disk images, regional and global climate data, and single-cell genomics, demonstrating its applicability across varied domains and data complexities.

---

> ### Author Rebuttal · Authors · 2025-07-31
>
> We thank the reviewer for their questions; we have grouped and will address them in order.
>
> **In short:** we clarify possible misunderstandings, provide new quantitative score‑alignment and intrinsic‑dimension results, and add stronger baselines on the MFEP task; together, these demonstrate that the theoretical findings transfer to applications.
>
> ### Global optimality
>
> While this is a relatively strong assumption, it common to many theoretic investigations of deep learning models. The goal is to explain the performance of DPA, which was already demonstrated in the original paper on complex data; in practice, our experiments show that even with approximate optima, one can observe the presented theoretical properties *and* that they lead to very desirable outcomes.
>
> For example, we provide **numeric results of the *cosine similarity*** between both sides of Eq. (7) - that is, the level set vector and the true data score - for the two examples in Fig. 1. The results were evaluated  on a 100 × 100 grid with points where the pdf exceeds 0.5 % of its maximum kept:
>
> | data set         | latent component | mean  cosine | std        | 95 %‑ile | points kept |
> | ---------------- | ---------------- | ------------ | ---------- | -------- | ----------- |
> | Standard Normal  | 0                | 1.000000     | 0.000000   | 1.000000 | 5 088       |
> | Standard Normal  | 1                | 1.000000     | 0.000000   | 1.000000 | 5 088       |
> | Gaussian Mixture | 0                | 1.000000     | 3.1 × 10⁻⁸ | 1.000000 | 4 729       |
> | Gaussian Mixture | 1                | 1.000000     | 3.0 × 10⁻⁸ | 1.000000 | 4 729       |
>
> Similarly, The conditional independence results were presented in **Sec. 1.2. of the supplement**  (`supplement.pdf` located at the root of the provided archive). *Suppl. §1.2, Tab. 2-3*  show for the **noisy** S-curve dataset that the *conditional mutual information*  and the *conditional distance correlation* indicate conditional independence even with the presence of noise and imperfect training.
>
> Furthermore, the stability of the results (across numerous random seeds) presented in Sec. 1.1 of the supplement indicate that finding a *good* optimum that displays the score alignment property is *not at all difficult* for DPA. We add further comparisons to two more advance autoencoder models: - $\beta-VAE$ (Higgins et al. 2017) and  $\beta-TCVAE$ (Chen et al. 2018) below as an extension to Tab 1. in the Supplement:
>
> | Model         | Param. comp.$\bar z_{\parallel}$ | Chamfer $\bar{d}_{\mathrm C}$ | Hausdorff $\bar{d}_{\mathrm H}$ | Mean 95th-perc. error | $\bar{d}_{\text{MFEP}\rightarrow\text{path}}$ | $\bar{d}_{\text{path}\rightarrow\text{MFEP}}$ |
> | ------------- | -------------------------------- | ----------------------------- | ------------------------------- | --------------------- | --------------------------------------------- | --------------------------------------------- |
> | $\beta$-VAE   | 0.5 ± 0.51                       | 0.450 ± 0.288                 | 1.172 ± 0.512                   | 1.051 ± 0.477         | 0.539 ± 0.180                                 | 0.360 ± 0.462                                 |
> | $\beta$-TCVAE | 0.375 ± 0.49                     | 0.377 ± 0.077                 | 1.378 ± 0.501                   | 1.228 ± 0.433         | 0.591 ± 0.180                                 | **0.164 ± 0.101**                             |
>
> We will consider moving some of these results to the main paper in the final version, and are performing further experiments.
> In short, all experiments point to the *observability of the derived theoretical properties in practice*.
>
> ### Manifold Parameterizability Assumption
>
> **Encoder capacity:**
> > “…the encoder function class must be sufficiently expressive to realize an exact parameterization of the underlying data manifold, which may not hold …”
>
> We apologize for any confusion—our Theorem 3.4 does **not** assume the encoder can already parameterize the manifold **exactly**.
> **Definition 3.1** introduces *K′‑parameterizability* **of the data manifold**, not of the network. It states only that there exists some latent width K′ at which the **best attainable (nested) loss term stops decreasing**. This is always the case for any finite-dimensional manifolds, with $K=p$ in the "worst" case.
>
> **Exact manifold maps are merely a special case:** *Corollary 3.5* shows that **if** an encoder exactly parameterizes a K‑manifold **and** is globally optimal among K‑latent models, then the K dimensions are *automatically* K′‑best with K′=K. This is a *sufficient* but **not necessary** scenario. *Theorem 3.4* *still applies* whenever a K′‑best-approximating encoder exists—even if $L_{K′}^*$ is positive; that is if:
> 	1. the data are *noisy* and the  cannot be learned as an exact manifold
> 	2. The manifold can only be approximately learned due to its *complexity*
> Then, the latents beyond $K'$ capture, at most, repetitions of that irreducible error.
>
> We wish to thank the reviewer for prompting this clarification and will add a sentence further distinguishing "*K′‑parameterizable data*" (a definition) from *K'-best-approximating* encoder (a requirement for Thm. 3.4) and the *exactly-parameterizing* encoder (Cor. 3.5, a special case).
>
> Thus, in practice for a very complex data, DPA might use slightly more latents $K' > K$ to "explain" the noise (or complexity); however, past $K'$ the latents will still be uniformative. It is unlikely that any other dimensionality reduction technique would perform much better in that case as the limitation is on the neural network as such.
>
>
> ### Generalization to More Complex data
>
> All theoretical results presented hold for **any** intrinsic/ambient dimensionality combination. In order to provide visual intuition, we opted for simple examples with a known data density (and hence the score). For example, the Muller-Brown example is an example with a known score that nonetheless demonstrates a large practical impact of the findings, as the method could be used to significantly speed up simulations.
>
> The scalability question is largely resolved by **prior empirical work**:  the original DPA paper already includes results for e.g.,  MNIST, climate data, etc., demonstrating competitive reconstruction and an automatically inferred latent dimension of $\approx$ 32 for MNIST ([17], Fig. 7). Our contribution is to **explain why** the method performs so well. For example, the global precipitation field results ([17], Fig. 9), which represent the periodic `month` dimension in a polar representation can be explained by our results (cf. lines 212-219).
>
>
> ### Finite Samples & Imperfect Training
>
> We believe that the experiments presented already demonstrate robustness to finite sample sizes and imperfect training.
>
> For a more theoretical guarantee:
> 1. The DPA decoder is an *Engression* network. The original paper provides explicit finite‑sample error bounds (Theorem 3, Shen & Meinshausen 2024).
> 2. The DPA encoder is a generic neural network, so standard results apply there.
>
> ### Full-Row-Rank Assumption
>
> 1. This assumption is **only invoked for Theorem 2.6**
>
> 2. If it's violated for a specific level set, the Theorem is simply **silent** on that specific level set; that is, this is not a pre-condition of the theorem.
>    Further discussion is provided in lines 108-112.
> 3. In our experiments, we observe level sets that are well-behaved for models whose training has converged (cf. the above alignment result).

---

> > ### Comment · Reviewer_YCG7 · 2025-08-02
> >
> > We thank the authors for their detailed and thoughtful response. The discussion on global optimality and manifold parameterizability was especially helpful in clarifying my earlier misunderstanding regarding the assumptions behind Theorem 3.4 and Corollary 3.5.
> >
> > I will raise my score accordingly.

---

> > > ### Author Response · Authors · 2025-08-03
> > >
> > > Thank you for your quick and positive follow‑up and for taking the time to reconsider our submission. We’re glad the clarifications were helpful and appreciate the discussion and your revised assessment!

---

### Official Review · Reviewer_NE6s · 2025-06-28

**Clarity:** 3
**Significance:** 2
**Originality:** 3
**Rating:** 5
**Confidence:** 4

**Summary:**

In this submission, the author(s) investigate the theoretical properties of Distributional Principal Autoencoder (DPA): (1) They establish a closed-form relationship connecting optimal level-set geometry to the data-distribution score; (2) For data residing on a manifold approximable by the encoder, encoder components exceeding the manifold's dimensionality (or its optimal approximation) exhibit conditional independence from the data, thus conveying no supplementary information.

**Questions:**

1. Can the author(s) discuss the potential impact of the theoretical results?
2. Can the author(s) provide some additional numerical results on the intrinsic dimension estimation? This is an important contribution of the work but there are no numerical demonstration on this point.
3. Can the author(s) provide some discussions on their assumptions and how they are related to the existing literature? How strong they are?
4. Can the author(s) discuss how the dimensionailty impacts the theoretical results? This is related to Q1. Because to my best, I note that many other generative learning techniques e.g. diffusion models performs quite well in the high-dimensional cases, and currently it is still unclear how the DPA performs in high-dimensional data setup.

**Ethical Concerns:**

["NO or VERY MINOR ethics concerns only"]

**Final Justification:**

The author(s) have addressed all my concerns and I raise my score accordingly.

**Limitations:**

See my comments on the weaknesses and questions.

**Paper Formatting Concerns:**

I think this submission does not have the formatting problem.

**Quality:**

3

**Strengths And Weaknesses:**

Strengths: The theoretical analysis is thorough and solid. The connection between DPA and the data-distribution score is interesting. At least for me, I haven't seen the similar result somewhere else.

Weaknesses: It is unclear how the theory impacts the community. Although the author(s) provide an application of the link between DPA and the data-distribution score in the case of Boltzmann distribution, it is unclear why this application matters. In addition, it is unclear the performance of the intrinsic dimension estimation, which is related to the second contribution of the work.

---

> ### Author Rebuttal · Authors · 2025-07-31
>
> We thank the reviewer for their interest and informative feedback. We have grouped and will address each aspect of it in order below.
> ### Impact
>
> Most approaches to unsupervised learning **trade-off data distribution approximation with dimensionality reduction** (cf. lines 321-326). Here, we show that a single method *provably* achieves **both** *without* any targeted regularization, approaching PCA in the second aspect *while* being a flexible generative model. The empirical findings (some additional ones presented here) further demonstrate that the theoretical findings can be observed even in practical experiments.
>
> While the use of the Energy score in deep learning is somewhat novel, similar models have been picking up considerable interest; for examples, see reference [20], De Bortoli et al. 2025 - “Distributional Diffusion Models with Scoring Rules”, and Shen and Meinshausen, 2024 -  “Engression".
>
> As a practical example: in molecular simulations, one typically wishes to find a compressed representation that can approximate the important, but *rare* transitions between physical states. Ideally, such a representation would parameterize the minimum free-energy path (MFEP), as traveling along the latter is the least-costly way for the system to undergo transitions. This representation can then be used to guide simulations. Because the MFEP is defined by the **Stein score of the Boltzmann density**, the fact that the latents align with the score lets DPA recover the reaction pathway **in one pass**, whereas current approaches need expensive iterative refinement (lines 244-252). This is further illustrated  in Sec. 1.1 of the Supplement (`supplement.pdf` in the provided archive). We furthermore **add more advanced autoencoder types** - $\beta-VAE$ (Higgins et al. 2017) and  $\beta-TCVAE$ (Chen et al. 2018) to *Tab 1.* below and demonstrate that DPA's score alignment is *provably* uniquely powerful for this scientifically important application:
>
> | Model         | Param. comp.$\bar z_{\parallel}$ | Chamfer $\bar{d}_{\mathrm C}$ | Hausdorff $\bar{d}_{\mathrm H}$ | Mean 95th-perc. error | $\bar{d}_{\text{MFEP}\rightarrow\text{path}}$ | $\bar{d}_{\text{path}\rightarrow\text{MFEP}}$ |
> | ------------- | -------------------------------- | ----------------------------- | ------------------------------- | --------------------- | --------------------------------------------- | --------------------------------------------- |
> | $\beta$-VAE   | 0.5 ± 0.51                       | 0.450 ± 0.288                 | 1.172 ± 0.512                   | 1.051 ± 0.477         | 0.539 ± 0.180                                 | 0.360 ± 0.462                                 |
> | $\beta$-TCVAE | 0.375 ± 0.49                     | 0.377 ± 0.077                 | 1.378 ± 0.501                   | 1.228 ± 0.433         | 0.591 ± 0.180                                 | **0.164 ± 0.101**                             |
>
> ### Empirical Results on Intrinsic Dimension
>
> In case this was missed, we'd like to point the reviewer to  **Sec. 1.2. of the supplement**  (`supplement.pdf` located at the root of the provided archive), which empirically evaluates the intrinsic dimension results. *Suppl. §1.2, Tab. 2‑3* thus show for the **noisy** S-curve dataset that the *conditional mutual information* `I(X;Z3|Z12)=0.13 bits` and the *conditional distance correlation* `dCorr=0.07`, meaning that the extraneous latent is effectively independent.
>
> With DPA, for many manifolds the uninformative latents (denoted as $U$) become (in practice) a *deterministic* function of the informative ones ($Z$), and satisfy Thm 3.4 directly. Then the results can be verified using the following estimators:
>
> | Estimator        | Idea                                                                                                                       |
> | ---------------- | -------------------------------------------------------------------------------------------------------------------------- |
> | **R²**           | Fit a non-linear regressor $U\approx g(Z)$ ; if $U=g(Z)$ deterministically the fit should explain ≈ 100 % of the variance. |
> | **ID‑drop**      | Compare intrinsic dimension of $(Z,U)$ vs. $Z$ via the Levina-Bickel MLE; $\approx$ 0 implies no extra d.o.f. in $U$.      |
> | **$H(U\mid Z)$** | Conditional entropy; large negative implies that $U$ tightly concentrated given $Z$.                                       |
>
> The results for various datasets are presented below:
>
> | Dataset                   | Description                                               | $R^2$  | ID‑drop \[2.5 %, 50 %, 97.5 %\] | $H(U\mid Z)$ \[nats\] |
> | ------------------------- | --------------------------------------------------------- | :----: | :-----------------------------: | :-------------------: |
> | Gaussian line             | 1-D line in $\mathbb{R}^2$ with orthogonal Gaussian noise | 0.9997 |  0.0112 / **0.0122** / 0.0132   |        −7.259         |
> | Parabola                  | $(t,t^{2})$                                               | 0.9997 |  0.0046 / **0.0048** / 0.0051   |        −9.190         |
> | Exponential               | $(t, exp(t))$                                             | 0.9996 |  0.0045 / **0.0047** / 0.0050   |        −9.162         |
> | Helix slice               | $(\cos t,\sin ⁡t, t)$                                     | 0.9995 |  0.0041 / **0.0043** / 0.0045   |        −6.090         |
> | Grid sum                  | $z = x + y$                                               | 0.9986 |  0.0023 / **0.0029** / 0.0034   |        −2.759         |
> | S-curve                   | from `sklearn`                                            | 0.9996 | −0.0031 / **−0.0014** / 0.0000  |        −1.762         |
> | Swiss-roll (3-D)          | from `sklearn`                                            | 0.9872 | −0.0048 / **−0.0025** / −0.0003 |        −0.042         |
> | S‑curve **($\beta$ = 2)** | ibid                                                      | 0.999  |     0.0030/ 0.0034/ 0.0037      |        ‑2.600         |
>
> One can also test Thm. 3.4 directly. For the Gaussian line example, we use a **Conditional randomization test**; under the null, we expect the p-values (100 replications) to be uniform. We confirm this with a K-S test:
>
> | K-S statistic $D$ | K-S p-value | p < 0.05 |
> | ----------------- | ----------- | -------- |
> | **0.061**         | **0.822**   | **4 %**  |
>
> We will consider moving some of these results to the main paper in the final version, and are performing further experiments.
>
> ### Assumptions
>
> The assumptions we make can be summarized as follows:
>
> - **Global optimality of encoder/decoder:** while this is a relatively strong assumption, it is not uncommon in theoretic treatments of  deep learning models.  Unlike, e.g., ref. [1], our results are **non-asymptotic and exact** given the optimum (lines 272-285). The analysis of optimal solutions can reveal important insights, even if in practice one can only approximate them. Furthermore, our empirical experiments corroborate the theoretical findings despite only approximate optimality of the models.
>
>   For example, we provide **numeric results of the *cosine similarity*** between both sides of Eq. (7) - that is, the level set vector and the true data score - for the two examples in Fig. 1. The results were evaluated  on a 100 × 100 grid with points where the pdf exceeds 0.5 % of its maximum kept:
>
> | data set         | latent component | mean  cosine | std        | 95 %‑ile | points kept |
> | ---------------- | ---------------- | ------------ | ---------- | -------- | ----------- |
> | Standard Normal  | 0                | 1.000000     | 0.000000   | 1.000000 | 5 088       |
> | Standard Normal  | 1                | 1.000000     | 0.000000   | 1.000000 | 5 088       |
> | Gaussian Mixture | 0                | 1.000000     | 3.1 × 10⁻⁸ | 1.000000 | 4 729       |
> | Gaussian Mixture | 1                | 1.000000     | 3.0 × 10⁻⁸ | 1.000000 | 4 729       |
> - **Encoder’s function class is sufficiently expressive:** We assume the encoder can approximate the data manifold. This is akin to a **universal approximation** assumption restricted to the manifold structure. This is not unheard of, and has been empirically evaluated (lines 191-193, [5]).
>
> - **Data lies on (or near) a low-dimensional manifold:** This is a common assumption to all representation learning. If this is not the case, Theorem 3.4 does not apply, but then again such a problem is not suitable for dimensionality reduction techniques. This does not impact Sec. 2.
>
> - **Jacobian full row rank almost everywhere on the level set:** We would like to point out that:
>   1. This only applies to Thm. 2.6
>   2. If it's violated for a specific level set, the Theorem is simply **silent** on that specific level set; that is, this is not a pre-condition of the theorem.
>    Further discussion is provided in lines 108-112; in our experiments, we observe level sets that are well-behaved for models whose training has converged (cf. the above alignment result).
>
>
>
> ### Data Dimensionality and High-Dimensional Performance
>
>
> All theoretical results presented hold for **any** intrinsic/ambient dimensionality combination. In order to provide visual intuition, we opted for simple examples with a known data density (in case of the score alignment).
>
> The original DPA paper already includes high-dimensional and complex data examples (e.g., MNIST, climate data) that already demonstrate DPA's strong *empirical* performance; our analysis *explains* it.
>
> In ordinary diffusion models, the latent dimension of the process is equal to the ambient one. While they learn the score, DPA thus additionally learns a *compressed representation*, for which we prove is *maximally informative* in case the data *can* be compressed. Thus we show that the method combines two desirable properties that are not found together in other models.

---

> ### Comment · Reviewer_NE6s · 2025-08-02
>
> Thanks for the detailed responses from the authors and they indeed addressed my questions. Hence I will raise my score. Good job!

---

> > ### Author Response · Authors · 2025-08-03
> >
> > Thank you for the quick response and the interest shown in the work. We are glad we were able to clarify the open questions and are thankful for your updated assessment.

---

> > > ### Author Response · Authors · 2025-08-07
> > >
> > > Thank you again for letting us know that our clarifications resolved your concerns. We noticed the system still shows the “Mandatory Acknowledgement” as pending; in case there is anything else we can help clarify before the Aug 8 deadline, please let us know.

---

### Official Review · Reviewer_kCFt · 2025-07-02

**Clarity:** 4
**Significance:** 3
**Originality:** 4
**Rating:** 6
**Confidence:** 3

**Summary:**

The paper presented two theoretical results regarding the Distributional Principal Autoencoder (DPA). First, a closed-form relation linking each optimal level-set geometry to the data-distribution score explains, to some extent,  DPA’s empirical ability to disentangle factors of variation of the data. Second, the paper proved that if the data lies on a manifold that can be approximated by the encoder, latent components beyond the manifold dimension carries no additional information. The above two theoretical results are helpful in deepening the understanding of DPA.

**Questions:**

Can the authors provide a proof for the second result based on a more general definition of K'-parameterizability? As an example, is it possible to prove that data with an intrinsic dimension of K' (and satisfies a set of loose conditions) can always be effectively learned by DPA using K' latent parameters? This kind of results will significantly improve the applicability of Result Two.

**Ethical Concerns:**

["NO or VERY MINOR ethics concerns only"]

**Final Justification:**

I maintain my previous assessment.

**Quality:**

4

**Strengths And Weaknesses:**

Strengths: The results of the paper may arouse theoretical interests and are mathematics-inclined.

Weakness: The generalization of the second major result in the paper relies on the definition of K'-parameterizability. However, the definition of  K'-parameterizability depends on DPA itself. That is to say, the K'-parameterizability here refers to the K'-parameterizability of DPA. This definition method obviously weakens the significance of this theoretical result.

---

> ### Author Rebuttal · Authors · 2025-07-31
>
> We thank the reviewer for the attentive review and for pointing out a possible generalization of our results.
> Below we clarify what is already covered and what remains open.
>
> 1. **K′‑parameterizable manifold always exists**: For every finite-dimensional dataset there is some smallest index $K′$ where $L^*_{k}$ stops decreasing; that K′ is “parameterizable” in Def. 3.1 with the trivial case  $K′=p$ . This is not necessarily connected to DPA per-se.
>
> 2. **K′‑best‑approximating encoder (Def. 3.3).** An encoder is **K′‑best-approximating** if
> 	1. it achieves the global optimum among all K′‑latent models *and*
> 	2. its *last* reconstruction term $L_{K′}[e,d]$ equals the optimal $L_{K′}^*$ .
>
>        When those two conditions hold, adding any surplus latent cannot lower the joint objective; strict propriety of the energy score then forces the surplus latents’ ORD to *repeat*  the optimal K′ ORDs, yielding the conditional‑independence conclusion in Theorem 3.4. Thus the applicability of Thm. 3.4 is connected to the **existence** of a **K′‑best-approximating** encoder.
>
> 3.  **Exactly parameterizable manifolds (Corollary 3.5).** If the data lie on a K‑manifold and an encoder $e_{1:K}$ *exactly* parameterizes it, then $L_K[e,d]=0$ . Therefore any global K‑latent optimal encoder/decoder is automatically **K‑best-approximating** and the existence of a K′‑best-approximating encoder  is reduced to the optimality of the parameterizing encoder in this case. The existence of the parameterizing encoder is not too stringent of a requirement (cf. lines 191-193), and its optimality is explored in Remark 3.6.
>
> 4. A **further generalization** could thus require that for a certain class of $K$-manifolds and a certain class of neural networks used for DPA, there exists a a network achieving $L_{K}[e,d] \approx 0$ . This would extend Cor. 3.5 and get around the existence issue, and is an open but feasible direction.
>
>
>  *In short,* Theorem 3.4 already applies whenever a K′‑best-approximating encoder exists; Corol. 3.5 simplifies the latter under the clean “exact manifold‑mapping” scenario, and we think universal‑approximation approaches can be used to tackle the more general case.

---

### Official Review · Reviewer_Dpaa · 2025-07-03

**Clarity:** 3
**Significance:** 2
**Originality:** 3
**Rating:** 4
**Confidence:** 2

**Summary:**

The paper analyzes the model called Distributional Principal Autoencoder (DPA), a recently proposed method which performs distribution matching using the energy score. The main contributions are showing that optimal encoders exhibit certain structure depending on the data distribution (Lemma 2.5), and the extra dimensions of an optimal encoder for DPA beyond the dimensionality of the manifold do not carry extra information (they are noise, Theorem 3.4) for \beta < 2 in the DPA. In particular, it is shown that for \beta = 2, the optimal encoder for DPA has level sets that align with the Stein score of the data distribution (Theorem 2.6), and numerical examples are given for this \beta = 2 regime.

**Questions:**

1. How do we interpret Equation (6) for values of \beta other than 2?
2. Are there properties similar to those established in Section 3 when \beta = 2?
3. The connections to dimensionality estimation are interesting, do you have any empirical validation for it?

**Ethical Concerns:**

["NO or VERY MINOR ethics concerns only"]

**Final Justification:**

The author clarified my main question in their response, and the authors responses to other comments helped clarify a lot of the confusion I had with the work.

**Limitations:**

The main limitation is that the theoretical results rely on finding the optimal encoder, which is not well-defined for \beta = 2, and is likely not computationally tractable for most problems. This is a common limitation in machine learning theory papers. The numerical examples support the overall hypothesis of the paper, but there are no numerical examples for the manifold results.

**Quality:**

3

**Strengths And Weaknesses:**

Strengths:
1) The theoretical analysis is novel and seems correct (though I did not have time to check the details of all proofs).
2) The result that DPAs can reveal the intrinsic dimension is fascinating.
3) Connection to the Stein score function can be useful and connects this paper to other research on distribution estimation.

Weaknesses:
1) The is heavy mathematical notation that I found difficult to follow at first, especially since I was not familiar with DPA. This is a lesser known model in this community at the moment, and readers may not be familiar with the topic and appreciate the analysis. This is the main reason for the lower score.
2) The numerical examples provided were low dimensional and simplistic. and the connection with score alignment was hard to see. Only comparisons to simple autoencoders and VAEs were given. Details on the setup for the experiments were not included in the paper (though code was included in supplementary material).

---

> ### Author Rebuttal · Authors · 2025-07-31
>
> We thank the reviewer for their careful reading and questions; we have grouped and will address each concern in order below.
> **In short:** we clarify notation, $\beta$‑regimes, and provide new quantitative score‑alignment and intrinsic‑dimension numbers; and add stronger baselines on the MFEP task.
>
> ### Clarity and significance
>
> We understand that the notation can be a bit heavy and would like to point the reviewer to Appendix A.1, which contains a summary of the notation. We will also expand the introduction to DPA (Sec. 2.1) in the final work. The gist of it is that the DPA is an autoencoder trained with an **energy score** loss (Eq. (10)) that ensures *distributionally correct reconstructions*, so all data points that were encoded with the same encoding value are *guaranteed* to be distributed under this conditional of the *original data distribution* (the *ORD*) on reconstruction.
>
> While the use of the Energy score in deep learning is somewhat novel, similar models have been picking up considerable interest; for examples, see reference [20], De Bortoli et al. 2025 - “Distributional Diffusion Models with Scoring Rules”, and Shen and Meinshausen, 2024 -  “Engression".  From a more general perspective, most unsupervised learning techniques **trade-off data distribution approximation with dimensionality reduction** (cf. lines 321-326). Here, we show that a single method *provably* achieves **both** *without* any targeted regularization, approaching PCA in the second aspect *while* being a flexible *non-linear* method.
>
> As a practical example: because MFEP is defined by the **Stein score of the Boltzmann density**, a single latent that aligns with the score lets DPA recover the reaction pathway **in one pass**, whereas current approaches need expensive iterative refinement (lines 244-252).
>
>
> ### Encoder optimality & $\beta$ Regimes
>
> We'd like to clarify that the optimal **encoder** is well-defined for $\beta=2$ (lines `102-107`). $\beta=2$ might induce the **only the decoder** to have a *non-unique* optimum, which would mean that the ORD is not a conditional of the original data distribution. This possible degeneracy has not been observed in any of the experiments performed.
>
>
> As for the general case of non-optimality of neural networks due to training, we'd again like to point to the experiments as even approximately optimal (by necessity) encoder/decoder pairs used seem to validate the theoretical results. Furthermore, we believe that understanding the mechanism behind a method is valid even if the exact optimum might not be achievable with finite data/training time.
>
> For a concrete illustration, we provide  **numeric results of the *cosine similarity*** between both sides of Eq. (7) - that is, the level set vector and the true data score - for the two examples in Fig. 1. The results were evaluated  on a 100 × 100 grid with points where the pdf exceeds 0.5 % of its maximum kept:
>
> | data set         | latent component | mean  cosine | std        | 95 %‑ile | points kept |
> | ---------------- | ---------------- | ------------ | ---------- | -------- | ----------- |
> | Standard Normal  | 0                | 1.000000     | 0.000000   | 1.000000 | 5 088       |
> | Standard Normal  | 1                | 1.000000     | 0.000000   | 1.000000 | 5 088       |
> | Gaussian Mixture | 0                | 1.000000     | 3.1 × 10⁻⁸ | 1.000000 | 4 729       |
> | Gaussian Mixture | 1                | 1.000000     | 3.0 × 10⁻⁸ | 1.000000 | 4 729       |
>
>
> For the questions concerning $\beta$:
> 1. A general way to summarize Eq. (6) would be that a general push/pull relation holds (in expectation) that matches an expected "flux" (as it is an integral of a divergence) $J_\beta$  on the LHS to the scaled variance of the level set on the RHS. The terms $J_\beta$ and $S$ represent a "push" from the data density to deform the level set, while the "variance" term on the RHS "pulls" it closer as to minimize the level-set variance.
>
> 2. We have performed additional experiments for the conditional independence result using $\beta=2$ (next section) and find that the Theorem *seems to hold in practice*. The requirement for $\beta<2$ is a technical one for the proof, as the issue is the same discussed above with regards to the possible degeneracy of the optimal *decoder*.
>
>
> ### Intrinsic Dimension Results
>
> We'd like to point to **Sec. 1.2. of the supplement** which empirically demonstrates the validity of the intrinsic dimension result. The Supplement also explains the provided code. *Suppl. §1.2, Tab. 2‑3* thus show for the *noisy* S-curve dataset that the *conditional mutual information* `I(X;Z3|Z12)=0.13 bits` and the *conditional distance correlation* `dCorr=0.07`, meaning that the extraneous latent is effectively independent.
>
> For many manifolds, the uninformative latents (denoted as $U$) become a *deterministic* function of the informative ones ($Z$), and satisfy Thm 3.4 directly. We present the following estimators:
>
> | Estimator           | Idea                                                                                                                       |
> | ------------------- | -------------------------------------------------------------------------------------------------------------------------- |
> | **R²**              | Fit a non-linear regressor $U\approx g(Z)$ ; if $U=g(Z)$ deterministically the fit should explain ≈ 100 % of the variance. |
> | **ID‑drop**         | Compare intrinsic dimension of $(Z,U)$ vs. $Z$ via the Levina-Bickel MLE; $\approx$ 0 implies no extra d.o.f. in $U$.      |
> | **$H(U \mid Z)$** | Conditional entropy; large negative implies that $U$ tightly concentrated given $Z$.                                       |
>
> The results for various datasets are presented below:
>
> | Dataset                   | Description                                               | $R^2$  | ID‑drop \[2.5 %, 50 %, 97.5 %\] | $H(U \mid Z)$ \[nats\] |
> | ------------------------- | --------------------------------------------------------- | :----: | :-----------------------------: | :----------------------: |
> | Gaussian line             | 1-D line in $\mathbb{R}^2$ with orthogonal Gaussian noise | 0.9997 |  0.0112 / **0.0122** / 0.0132   |          −7.259          |
> | Parabola                  | $(t,t^{2})$                                               | 0.9997 |  0.0046 / **0.0048** / 0.0051   |          −9.190          |
> | Exponential               | $(t, exp(t))$                                             | 0.9996 |  0.0045 / **0.0047** / 0.0050   |          −9.162          |
> | Helix slice               | $(\cos t,\sin ⁡t, t)$                                     | 0.9995 |  0.0041 / **0.0043** / 0.0045   |          −6.090          |
> | Grid sum                  | $z = x + y$                                               | 0.9986 |  0.0023 / **0.0029** / 0.0034   |          −2.759          |
> | S-curve                   | from `sklearn`                                            | 0.9996 | −0.0031 / **−0.0014** / 0.0000  |          −1.762          |
> | Swiss-roll (3-D)          | from `sklearn`                                            | 0.9872 | −0.0048 / **−0.0025** / −0.0003 |          −0.042          |
> | S‑curve **($\beta$ = 2)** | ibid                                                      | 0.999  |     0.0030/ 0.0034/ 0.0037      |          ‑2.600          |
>
> Furthermore, for the Gaussian line we test Thm. 3.4 directly with a **Conditional randomization test**; under the null, we expect the p-values (100 replications) to be uniform. We confirm this with a K-S test:
>
> | K-S statistic $D$ | K-S p-value | p < 0.05 |
> | ----------------- | ----------- | -------- |
> | **0.061**         | **0.822**   | **4 %**  |
>
>
> ### Further Experimental validation
>
> First, we'd like to address the relatively simple examples used for the experiments. We wish to provide visual intuition and verification of the theoretical results, which, for score alignment, is only possible for 2D examples with a known data density. The original DPA paper already includes high-dimensional and complex data examples that demonstrate DPA's *empirical* performance; here, we provide the *why*.
>
> The score alignment is additionally illustrated in Sec. 1.1 of the Supplement. We  **add  more advanced autoencoder types** - $\beta-VAE$ (Higgins et al. 2017) and  $\beta-TCVAE$ (Chen et al. 2018) to *Tab 1.* below:
>
> | Model         | Param. comp.$\bar z_{\parallel}$ | Chamfer $\bar{d}_{\mathrm C}$ | Hausdorff $\bar{d}_{\mathrm H}$ | Mean 95th-perc. error | $\bar{d}_{\text{MFEP}\rightarrow\text{path}}$ | $\bar{d}_{\text{path}\rightarrow\text{MFEP}}$ |
> | ------------- | -------------------------------- | ----------------------------- | ------------------------------- | --------------------- | --------------------------------------------- | --------------------------------------------- |
> | $\beta$-VAE   | 0.5 ± 0.51                       | 0.450 ± 0.288                 | 1.172 ± 0.512                   | 1.051 ± 0.477         | 0.539 ± 0.180                                 | 0.360 ± 0.462                                 |
> | $\beta$-TCVAE | 0.375 ± 0.49                     | 0.377 ± 0.077                 | 1.378 ± 0.501                   | 1.228 ± 0.433         | 0.591 ± 0.180                                 | **0.164 ± 0.101**                             |

---

> > ### Comment · Reviewer_Dpaa · 2025-08-04
> >
> > Thank you for the excellent clarifying response. My confusion with the $\beta=2$ issue has been resolved - this helps a lot. I had a better understanding of the paper after spending more time reading your responses to me and other reviewers. I have raised my score accordingly as I now support this paper for acceptance.

---

> > > ### Author Response · Authors · 2025-08-04
> > >
> > > We wish to sincerely thank the reviewer for the extra time they took to revisit the paper and the responses.
> > > We are glad that the questions have been resolved and appreciate the revised assessment!

---

### Note · Authors · 2025-08-12

Dear Area Chair and reviewers,

We'd like to take this opportunity to briefly summarize the discussions.

Of the points raised by the reviewers, most touched on the significance and the experimental validation.
In response, we have provided several additional experiments for both main results, showing that they can be readily observed in practice.
In terms of wider context, we have emphasized that this work *proves strong properties* for the data distribution learning and dimensionality reduction aspects of the DPA --- two aspects which are customarily considered a trade-off in unsupervised learning.
Furthermore, the work demonstrates that the first result has a direct impact when used in molecular dynamics simulations, an application of considerable scientific significance.


We wish to thank the reviewers for the suggestions. We will integrate the new experiments and clarifications, and further emphasize the impact in the final version.


Based on the discussions, our understanding is that all outstanding questions have been resolved.
We sincerely wish to thank you for the time and dedication in evaluating this work.


Best regards,
the Author(s)

---

### Decision · Program_Chairs · 2025-09-17

**Decision:**

Accept (poster)

**Comment:**

The paper studied the properties of distributional principal autoencoder (DPA) and provided solid theoretical analysis. The reviewers value the insightful theoretical results, but also raised questions regarding derivation, empirical evaluation and the practicality of the results. The authors provided additional results and clarified most of the concerns. The rebuttal successfully convinced two reviewers to raise their scores. Now, all reviewers share a positive view about the paper.  The AC agrees with the reviewers’ assessments and feels enthusiastic about the potential of the paper. The final version should incorporate the reviewers’ comments and also provide more clear clarifications as provided in the rebuttal.